# A comprehensive analysis of canonical biological pathways linking milk yield and quality traits to key fertility indicators in Murciano-Granadina dairy does

**María Pía Peláez Caro[1,2], Ander Arando Arbulu[1], José Manuel León Jurado[3], Juan Vicente Delgado Bermejo[1], Javier Fernández Álvarez[2], Francisco Javier Navas González** [1]*

**1** Department of Genetics, Faculty of Veterinary Medicine, University of Córdoba, Córdoba, Spain,
**2** CAPRIGRAN, National Association of Breeders of Murciano-Granadina Goat Breed, Granada, Spain,
**3** Centro Agropecuario Provincial de la Diputación de Córdoba, Córdoba, Spain

\* fjng87@hotmail.com

## Abstract

This study analyses relationships between milk yield, milk composition, and fertility in Murciano-Granadina dairy goats to test whether these associations reflect biological pathways underlying a production–fertility trade-off. Linear and non-linear patterns were examined using canonical discriminant analysis (CDA), regularized canonical correlation analysis (rCCA), and CHAID decision trees. The dataset included 32,693 artificial inseminations from 21,757 does and 29,390 milk records from 21,541 does; fertility was assessed using three indicators accounting for insemination timing and semen type. Mean fertility differed by semen preservation method, with fresh/chilled semen showing 18–20 percentage points higher fertility than frozen/thawed semen. Milk yield showed wide variability (8.7–6704.6 kg; mean 686.4 kg), whereas milk composition was relatively stable. After multicollinearity control (VIF > 178 to <5), CDA and rCCA revealed a low-dimensional structure dominated by a first canonical function, explaining 80.8–88.7% of discriminant variance and 97.3% of shared covariance. This dominant axis reflected an energy balance and metabolic partitioning pathway, with higher milk yield and solids concentration associated with reduced fertility; very high production (>2021 kg) consistently coincided with lower fertility, supporting a production–reproduction trade-off. A secondary axis represented an endocrine and lactation-timing pathway linking fertility to temporal variation in milk composition, while semen type constituted an autonomous semen-related pathway with the strongest standardized canonical coefficients. CHAID analysis identified non-linear relationships and biologically meaningful thresholds in milk traits, achieving ~44% classification accuracy and up to 87% reliability for high fertility. Optimal fertility was associated with intermediate ranges of milk yield and composition (protein 3.64–6.98%, fat 6.0–7.9%, dry matter 14–25%, lactose 5.6–6.5%, SCC < 5200 × 10³ cells/mL). Overall, milk yield

**Data availability statement:** Data is either included as a part of the paper or has been uploaded as supplementary information. Updated information can also be downloaded from the Spanish National System of Breed Information (ARCA) Public Repository. https:// www.mapa.gob.es/es/ganaderia/temas/zoo-tecnia/razas-ganaderas/razas/catalogo-razas/ caprino/murciano-granadina/.

**Funding:** Funding was not received for the development of the present study. The present research was carried out during the covering period of a Ramón y Cajal Post-Doctoral Contract with the reference MCIN/ AEI/10.13039/501100011033 and the European Union "NextGenerationEU"/PRTR. The funders had no role in study design, data collection and analysis, decision to publish, or preparation of the manuscript.

**Competing interests:** The authors have declared that no competing interests exist.

and composition showed statistically robust but moderate associations with fertility, consistent with reproductive biology multifactorial nature of and gaining relevance when interpreted jointly with metabolic, endocrine, and semen-related factors.

## Introduction

Genetic selection has been a major driver of progress in the dairy sector, traditionally emphasizing economically important traits related to productivity and profitability [1]. In parallel, longevity and functional traits have gained relevance in breeding programmes, particularly in the Genetic Improvement Program of the Murciano-Granadina goat managed by CAPRIGRAN [2]. This breed is now among the most important dairy goat populations worldwide due to its high milk quality—especially for cheese production—combined with adaptability to diverse environments and grazing systems [3,4].

Despite these advances, a persistent challenge in dairy breeding is the antagonistic relationship between milk production and female fertility. While artificial insemination (AI) is a cornerstone of genetic improvement [5], sustained selection pressure for milk yield has been associated with declining reproductive performance, as extensively documented in dairy cattle and increasingly reported in goats [6,7]. However, in goats—and particularly in Murciano-Granadina populations—it remains unclear whether fertility variation is linked to specific milk traits, to overall production intensity, or to broader physiological patterns integrating milk yield, milk composition, and reproductive function. This unresolved issue represents a key research gap with direct implications for breeding and management strategies.

Fertility is a complex, multifactorial trait influenced by genetic background, environmental conditions, management practices, and productive status [5,8,9]. One widely proposed explanatory framework for reduced fertility in high-producing females is negative energy balance, which arises when nutrient demands for lactation exceed intake [10]. Under such conditions, selection favoring productivity may be associated with compromised reproduction, health, or longevity [10,11]. In goats, associations have been reported between reproductive indicators and milk yield, fat content, litter size, and parity [7], although effect sizes are often moderate and context-dependent.

Milk composition traits may provide additional insight into these relationships, as they reflect aspects of metabolic and physiological status beyond total milk yield. Previous studies suggest that milk fat, protein, and lactose are associated with energy balance and nitrogen metabolism, which are in turn related to reproductive performance [12,13]. Importantly, these relationships should be interpreted as associative rather than causal, particularly in observational studies, as milk traits act as indicators of underlying physiological states rather than direct drivers of fertility outcomes.

Similarly, genetic variation affecting endocrine signaling, metabolism, and energy partitioning has been shown to influence both lactation and reproduction [14–16]. These shared influences are largely mediated through interconnected hormonal and metabolic networks rather than single genes with isolated effects [17–19].

Consequently, relationships between milk traits and fertility are best conceptualized as emerging from integrated biological systems rather than from direct mechanistic pathways measurable at the phenotypic level.

Management strategies such as extending the interval between parturition and insemination or increasing hormonal stimulation have been proposed to mitigate fertility losses in high-producing goats [20,21]. However, these approaches are not universally applicable, particularly in prolific breeds such as Murciano-Granadina goats, where excessive hormonal intervention may increase reproductive risk and reduce efficiency [4,22]. Moreover, the effectiveness of hormonal treatments has been shown to decline with increasing production level [5], reinforcing the need for population-level analytical approaches.

In this context, multivariate methods offer advantages over univariate or mixed-model approaches by allowing simultaneous evaluation of multiple correlated traits. Canonical discriminant analysis (CDA) is particularly suited to identifying combinations of milk traits that discriminate between successful and unsuccessful inseminations, while canonical correlation analysis (CCA) captures shared multivariate structure between milk production and fertility traits. Rather than inferring causality, these methods enable detection of low-dimensional association structures that may reflect recurrent physiological or management-related processes operating at the population level.

The central hypothesis of this study is that associations between milk yield, milk composition, and fertility in Murciano-Granadina goats are structured around a limited number of recurrent biological association pathways, rather than independent effects of individual traits. These pathways are inferred from multivariate patterns and are interpreted as reflecting integrated processes related to energy balance, endocrine regulation, and semen-related management factors.

Accordingly, the aim of this study is to examine multivariate associations between milk yield, milk composition, and fertility in Murciano-Granadina goats using CDA and CCA, complemented by CHAID decision tree analysis, in order to identify recurrent biological association pathways linking dairy production and reproductive performance at the population level. These pathways are interpreted as integrated patterns rather than direct mechanistic causation and provide a biologically informed framework for understanding production–fertility relationships in this breed.

## Materials and methods

### Sample and study conditions

This 10-year longitudinal study (2010–2019) included 21,757 Murciano-Granadina does born between September 1999 and June 2018. A total of 32,693 artificial insemination records collected between January 2010 and December 2019 were analyzed. Semen was obtained from 115 Murciano-Granadina bucks belonging to the national breeding program and selected for proven fertility and high genetic merit (Fig 1). All female goats registered in the official breeding and reproductive recording system with artificial insemination data during the study period were evaluated under the program's standard inclusion and quality-control criteria. These criteria were applied prior to analysis, and no animals or records were excluded, as all data met the required standards for completeness, consistency, and biological plausibility. This study was retrospective and observational; no experimental or management interventions were applied, and all procedures followed routine farm and breeding-program practices.

### Doe location

Fig 2 shows the distribution of the Murciano-Granadina goat breed across Spain. S1 Table details the geographic distribution of participating farms, including farm acronym, province, town, and coordinates, supporting spatial analyses of environmental and management influences on reproductive performance.

### Buck location

The bucks (Fig 2) were housed at the Andalusian Goat Selection and Improvement Center in Albolote until 2014, after which they were moved to Fuente Vaqueros (Granada, Spain). Both locations are in the Vega de Granada and share a continental Mediterranean climate.

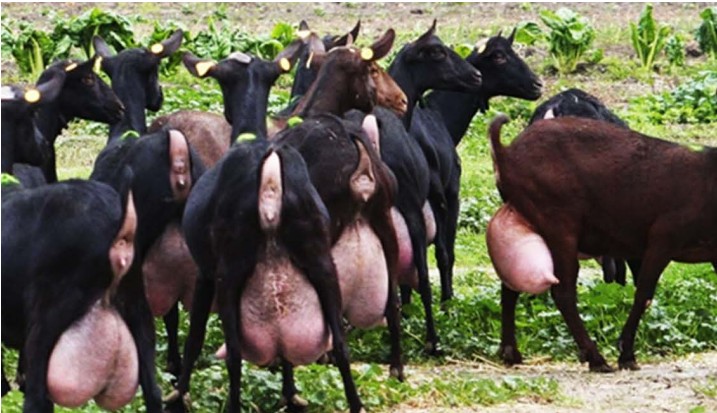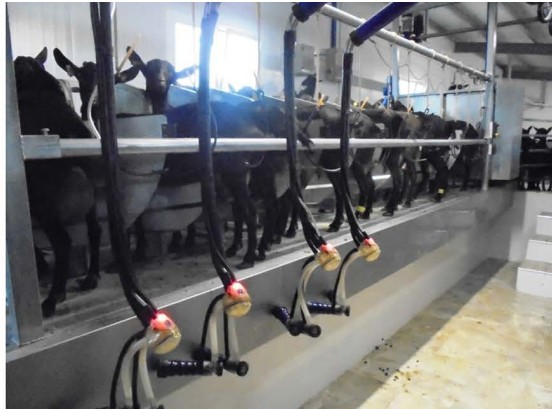

**Fig 1. Murciano-Granadina does on field and herd and in the milking room.**

The region shows marked diurnal temperature variation, with daily differences often exceeding 20 °C. Rainfall occurs mainly from late autumn to early spring, with 6.2–8.7 frost days per season and a mean annual precipitation of 419.5 mm. Throughout the study, bucks were managed under uniform feeding conditions, receiving 0.5 kg/day of commercial concentrate with ad libitum hay, water, and mineral blocks to minimize dietary variability.

## Semen preparation

Ejaculates were collected using an artificial vagina and immediately placed in a 37 °C water bath for evaluation. Samples were selected for artificial insemination (AI) when volume exceeded 0.5 mL, sperm concentration was $> 3,000 \times 10^6$ sperm/mL, and mass motility was $> 4$.

For chilled semen, INRA 96 (IMV Technologies, France) was used as extender, and sperm concentration was adjusted to $200 \times 10^6$ sperm per straw; straws were stored at 5 °C and used within 4–6 h. For cryopreservation, Triladyl (IMV Technologies, France) was used at a final concentration of $150 \times 10^6$ spermatozoa per straw. Freezing was performed using a Digitcool programmable cryo-freezer (IMV Technologies, France), and straws were thawed at 37 °C for 30 s before insemination.

Inseminations were carried out on 110 commercial Murciano-Granadina farms approximately 46 h after sponge removal using cervical insemination with an illuminated speculum. Pregnancy was diagnosed 42 days post-insemination by transabdominal ultrasonography (5 MHz probe), and kidding records were used to confirm progeny origin.

## Fertility evaluation

Three complementary fertility indicators were defined to capture distinct sources of variation associated with the female physiological response to artificial insemination (pregnancy outcome), environmental and management conditions, and semen-related factors. All indicators were calculated at the insemination-event level and assigned uniformly to all goats inseminated within the same event.

a) Fertility per day of insemination was defined as the proportion of inseminated goats that became pregnant among all females inseminated on the same day, irrespective of buck identity or semen type [23]. This indicator captures shared environmental, managerial, and temporal effects acting on reproductive performance on a given insemination day.

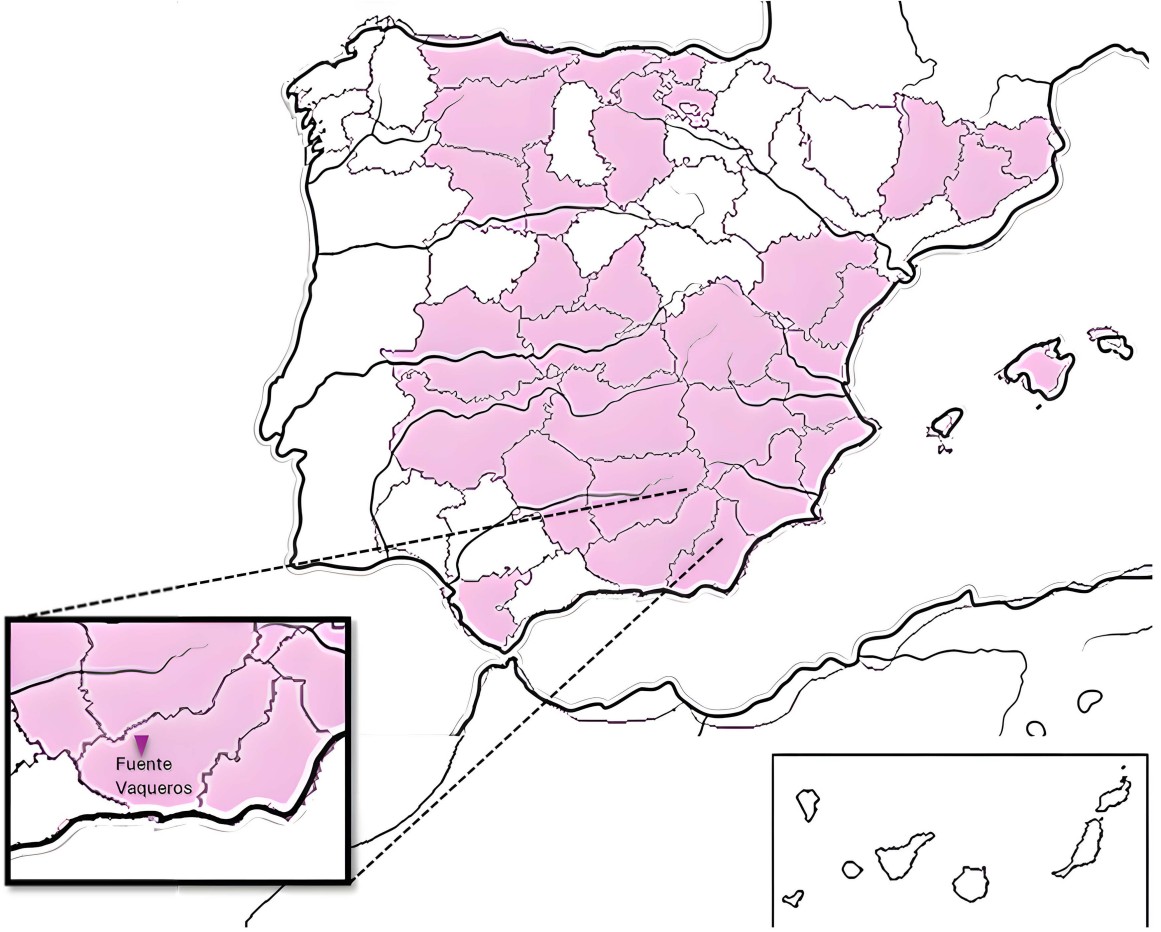

**Fig 2. Murciano-Granadina goat breed distribution across the Spanish territories and Location of Caprigen (Goat Biotechnology Center) in Fuente Vaqueros, Granada (Spain) (Navas González, 2026).** The map is an original illustration created by the author for this study, edited in Microsoft PowerPoint (Microsoft 365) and rendered using Paint.

b) Fertility per buck batch and day of insemination was defined as the proportion of inseminated goats that became pregnant among all females inseminated on the same day using semen from the same buck [24]. This metric reflects buck-specific fertility while controlling for daily environmental and management conditions.

c) Fertility by semen type (fresh/chilled vs. frozen/thawed) was defined as the proportion of inseminated goats that became pregnant among all females inseminated on the same day using the same semen preservation method [25]. This indicator isolates fertility differences attributable to semen processing and cryopreservation.

The definition of fertility categories was supported by exploratory frequency analysis of fertility percentages stratified by semen type, day of insemination, and buck batch (Fig 3). Across all stratifications, fertility values showed unimodal distributions that approached normality, with the majority of observations clustered around intermediate values and progressively fewer observations toward the lower and upper extremes.

Accordingly, fertility thresholds (≤20%, 20–40%, 40–60%, 60–80%, ≥80%) were defined to represent the lower tail, intermediate ranges, and upper tail of the empirical distributions. These cut-offs were therefore data-driven and reflect the observed structure of the fertility data rather than arbitrary or biologically imposed limits.

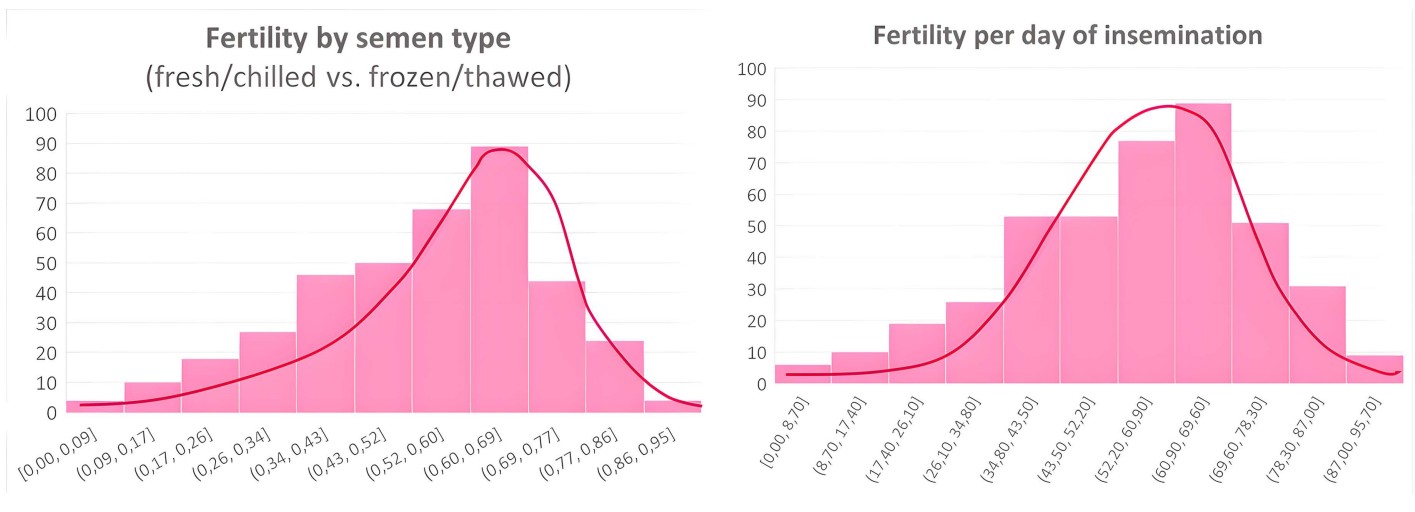

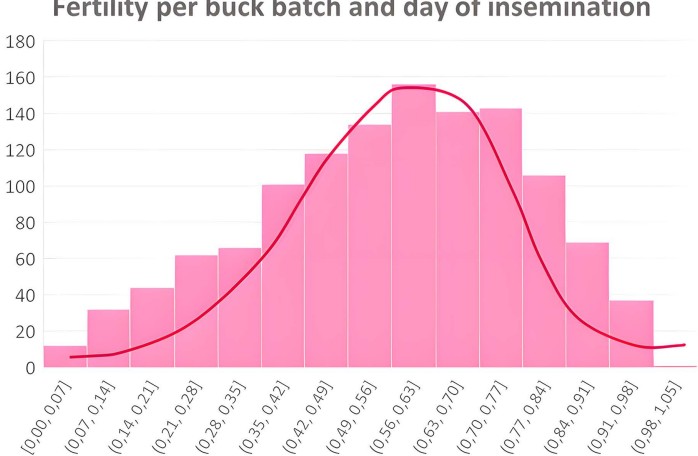

**Fig 3. Frequency distributions of fertility percentages by semen type (fresh/chilled vs. frozen/thawed), day of insemination, and buck batch combined with insemination day.** Histograms depict empirical fertility frequencies, with overlaid kernel density estimates and fitted Gaussian curves illustrating distributional trends. The unimodal shape and approximate symmetry of the distributions indicate fertility values approaching normality, supporting the definition of fertility categories based on empirical distribution tails and central ranges.

This distribution-based categorisation is consistent with fertility benchmarking practices commonly applied in goat artificial insemination studies, where fertility outcomes are interpreted relative to population-level performance and distributional properties rather than fixed biological thresholds [26,27].

## Milk yield and composition standardization

The Murciano-Granadina production system is characterized by two annual kidding seasons, with milking periods of approximately 210–240 days [28]. Total milk yield (g), milk composition (%), and somatic cell count (SCC × 10³ cells/mL) were estimated up to 210 days of lactation following [29], and composition percentages were converted into grams. Individual milk yield was calculated using the real yield ($RY_i$) formula:

$$RY_i = d_{1i}Y_{1i} + 30\sum_{i2} Y_{ij} + [d_{2i} - 30(n_i - 2)]Y_{ni},$$

where $Y_{1i}$ and $Y_{ni}$ represent milk yield at the first and last controls, respectively; $n_i$ is the number of controls; and $d_1$ and $d_2$ correspond to the intervals between kidding and the first control, and between the penultimate and last controls. On average, five milk controls per goat were performed in accordance with Royal Decree 368/2005 [30].

To account for variation in lactation length, milk yield was normalized to 150–305 days of lactation using parturition dates and milk control records. Normalized yield ($NY_i$) was calculated as:

$$NY_i = d_1 P_1 + A + B,$$

where A and B represent trapezoidal integrations between consecutive milk controls. Accumulated milk production from 150 to 305 days was computed as:

$$MP150 - 305 = \sum [\frac{pldc_i + pldc_{i+1}}{2} \times I_{i,i+1}].$$

Monthly milk samples were analyzed at the Official Milk Quality Laboratory (Córdoba, Spain). Fat, protein, dry matter, and lactose contents were determined using a MilkoScan™ FT1, and SCC × $10^3$ cells/mL using a Fossomatic™ FC. After data cleaning, 29,390 milk and composition records from 21,541 does born between September 1999 and June 2018 were retained for statistical analyses. Milk component yields (g) were calculated from normalized milk yield at 210 days of lactation.

Milk yield and composition traits were standardized at 150, 210, 240, and 305 days of lactation, as these points correspond to biologically and operationally meaningful stages of the lactation curve in dairy goats. Standardization at 150 days reflects early to mid lactation, near peak production and high metabolic demand, whereas 210 and 240 days represent mid-lactation phases characterized by greater physiological stability. The 305-day point corresponds to the internationally recognized reference length for a complete lactation and is routinely used in official performance recording and genetic evaluation. This approach is consistent with Spanish legislation governing Official Milk Recording, which mandates standardized records at 150 and 305 days to ensure comparability across animals and herds [30]. Inclusion of the intermediate stages (210 and 240 days) enables assessment of fertility–lactation relationships across distinct physiological phases rather than relying solely on cumulative lactation measures.

## Statistical analysis

The statistical analysis followed a sequential and hierarchical workflow (Fig 4). Exploratory analyses, including descriptive statistics, correlation analysis, and multicollinearity diagnostics, were first conducted to characterize data structure and motivate multivariate modeling. The central hypothesis—that milk production and fertility are structured around a limited number of recurrent biological pathways—was formally tested using regularized canonical correlation analysis (rCCA), which served as the primary inferential method. Canonical discriminant analysis (CDA) was subsequently applied to evaluate whether the same multivariate structure discriminated fertility categories based on milk yield and composition traits standardized at 150, 210, 240, and 305 days of lactation. CHAID decision trees were used as a complementary exploratory tool to investigate non-linear and threshold effects and to facilitate biological interpretation. Each analytical approach was validated using a method-specific cross-validation strategy appropriate to its inferential objective.

**Common A priori assumptions.** Canonical discriminant analysis (CDA) assumes linear relationships between predictors and group membership and requires several statistical conditions for valid inference [31,32]. Canonical correlation analysis (CCA) extends this framework by assessing multivariate associations between two sets of continuous variables through pairs of maximally correlated canonical variates [33,34]. Unlike CDA, CCA does not require predefined groups but assumes multivariate normality, low multicollinearity within variable sets, and adequate sample size. Canonical correlations (Rc) quantify shared variance between sets ($Rc^2$), and interpretation is based on coefficients, loadings, and

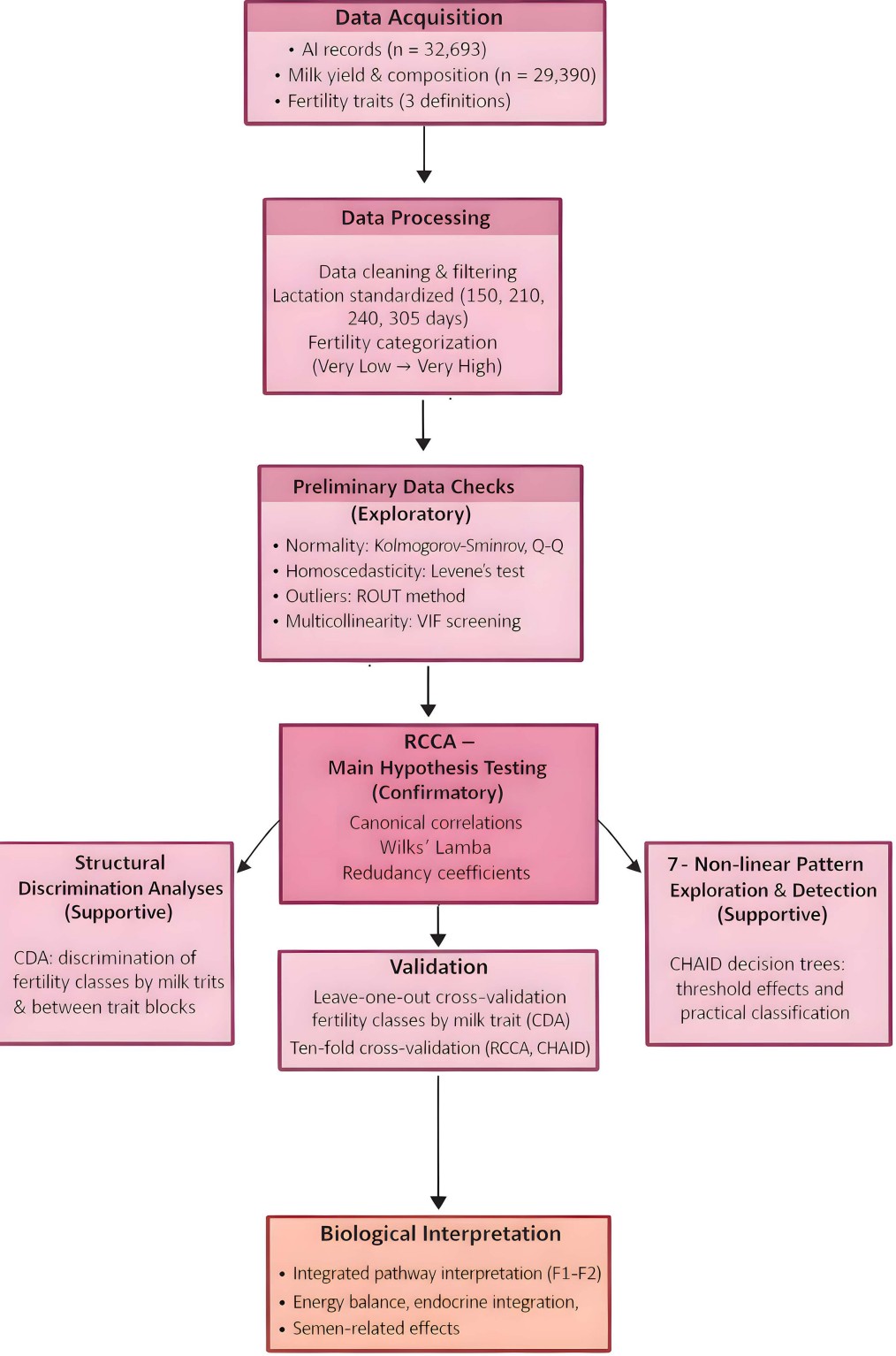

**Fig 4. Analytical workflow of the study.** Schematic overview of the analytical framework, from data acquisition and preprocessing through confirmatory multivariate analysis (rCCA), supportive analyses (CDA, CHAID), validation, and biological interpretation of fertility–lactation relationships.

redundancy indices. Together, CDA and CCA provide complementary perspectives on group discrimination and structural relationships among variables.

Prior to analysis, multivariate assumptions were evaluated. Normality was assessed using the Kolmogorov–Smirnov test in XLSTAT 2014 (Addinsoft, France) and Q–Q plots, indicating approximate normality. Homogeneity of variances was tested using Levene's test, revealing heteroscedasticity ($p < 0.05$). Outliers were screened using the ROUT method ($Q = 1\%$) in GraphPad Prism 9.0, with none detected. Sample adequacy was ensured by maintaining at least ten observations per predictor [35], and multicollinearity was controlled using variance inflation factors (VIF < 5) [36].

**Canonical discriminant analysis (CDA).** Canonical discriminant analysis (CDA) was implemented in XLSTAT (Addinsoft Pearson Edition 2014) following the default estimation framework. CDA was treated as a descriptive multivariate technique aimed at characterizing separation among predefined fertility classes through linear combinations of the selected predictors. In XLSTAT, discriminant functions are estimated using all available observations to compute group centroids and pooled within-group covariance matrices; no automatic data partitioning into training or testing subsets is applied by default. The number of canonical discriminant functions was inherently constrained by the number of fertility groups and by the rank of the covariance matrix, limiting model dimensionality and preventing overparameterization. Model adequacy and discrimination strength were subsequently evaluated using Wilks' lambda, Pillai's trace, Bartlett's test of sphericity, and squared Mahalanobis distances, as described below.

**Explanatory variables – Unstandardized and standardized milk yield and composition:** Canonical discriminant analyses (CDAs) were performed to design a tool for classifying milk yield and composition records (Unstandardized and Standardized fat, protein, lactose, dry matter and somatic cells, at 150, 210, 240, 305 days of lactation) assessing whether linear combinations of such could describe within- and between-fertility level clustering patterns (from very low, low, medium, high and very high).

**Clustering variables – Fertility indicators scales:** Clustering variables included fertility per day of insemination (daily percentage of successful pregnancies), fertility per buck batch and day of insemination, and fertility per day by semen type (fresh/chilled vs. frozen/thawed). All fertility traits were expressed as percentages and categorized based on normality distribution plots into five classes: very low (0–20%), low (20–40%), medium (40–60%), high (60–80%), and very high (80–100%), providing a standardized framework for evaluating fertility outcomes.

**Variable selection:** Variable selection was performed using regularized forward stepwise multinomial logistic regression. Priors were adjusted in SPSS v26.0 (SPSS Inc., Chicago, IL) to account for unequal sample sizes, minimizing potential bias. Both forward and backward methods identified the same variables, but forward selection was chosen for computational efficiency.

**Sample size considerations:** Sample adequacy followed established guidelines, requiring a minimum of 20 observations per 4–5 predictors, with independent variables limited to n–2. The ratio of observations to variables in this study exceeded the recommended minimum by a factor of 4–5, ensuring robust discriminant analyses [37].

**Multicollinearity analysis:** Multicollinearity was assessed before conducting Canonical Discriminant Analysis (CDA) to prevent inflated variance explanations. Variance inflation factors (VIF) and tolerance values were calculated, with VIF computed as:

$$\text{VIF} = 1/(1 - R^2)$$

where $R^2$ is the coefficient of determination from regressing a predictor on all other predictors. A VIF threshold of 5 was applied [36].

**Canonical correlation dimensions:** The maximum number of canonical correlations was determined by the number of variables in the smaller set. While the first canonical root explained most of the variance, all correlations ≥ 0.30 were considered meaningful, accounting for approximately 10% of the explained variance.

**CDA efficiency and model reliability:**   Bartlett's test of sphericity

This test examined whether the correlation matrix significantly differed from an identity matrix. A significant result ($p < 0.05$) indicated the suitability of the data for CDA.

Wilks' lambda

Wilks' lambda was used to evaluate discriminant power, with smaller values indicating stronger group separation. Significance was tested via $\chi^2$ approximation ($p < 0.05$).

Pillai's trace

Pillai's trace criterion was selected due to the unequal group sizes, providing robustness against violations of multivariate normality and homogeneity. A significance level of $\leq 0.05$ indicated that the predictors contributed significantly to discrimination.

Canonical coefficients, loadings, and spatial representation

Dimensionality reduction was first performed using Principal Component Analysis (PCA). Variables with discriminant loadings $\geq |0.40|$ were retained. Standardized canonical coefficients were interpreted, with larger coefficients indicating stronger discriminating power.

Squared Mahalanobis distances between groups were computed as:

$$D^2ij = (\overline{Y}_i - \overline{Y}_j) \times COV^{-1} \times (\overline{Y}_i - \overline{Y}_j)$$

where $D^2ij$ represents the squared Mahalanobis distance between groups i and j, $\overline{Y}_i$ and $\overline{Y}_j$ are the group mean vectors, and $COV^{-1}$ is the inverse of the covariance matrix. These distances were converted to Euclidean metrics and clustered using UPGMA dendrograms in MEGA X v10.0.5 (Pennsylvania State University, USA).

Discriminant function reliability and cross-validation

Leave-one-out cross-validation was used to estimate predictive accuracy. Classification reliability was evaluated using the hit ratio (proportion of correctly classified semen doses by moon phase) and leave-one-out cross-validation. Press's Q statistic was calculated as

$$Q = [(n - n'K)^2]/[n(K - 1)],$$

where n is the total number of observations, n' the number correctly classified, and K the number of groups. Values exceeding $\chi^2 = 6.63$ (df = 1, $p < 0.01$) indicated accuracy at least 25% above chance.

**Data mining with CHAID decision tree.**

**CHAID methodology:**   To complement the canonical discriminant analysis (CDA), a Chi-squared Automatic Interaction Detection (CHAID) decision tree was applied to classify and predict fertility level using discretely categorized fertility indicators. Tree construction relied on Pearson's chi-square tests for node splitting, with statistical significance evaluated at *p* < 0.05 and adjusted using Bonferroni correction to account for multiple testing.

The CHAID algorithm was implemented in XLSTAT (Addinsoft Pearson Edition 2014, Addinsoft, Paris, France). Model complexity was controlled exclusively through pre-pruning (early stopping) criteria, with no post-pruning procedures applied. The maximum tree depth was fixed at five levels. Node size constraints required a minimum of two observations in parent nodes and at least one observation in child nodes for a split to be considered.

Node splitting and category merging were based on Pearson's chi-square statistic, using a 5% significance threshold for both operations. Category redivision within nodes was permitted, allowing previously merged categories to be re-split when statistically justified. Both merge and split significance thresholds followed the default XLSTAT configuration.

For quantitative explanatory variables (Milk yield, composition and quality traits), discretization was performed prior to tree construction using univariate clustering into ten intervals. Model estimation was allowed to proceed for up to 1,000 iterations, with convergence defined by a minimum change threshold of 0.00001. The dependent variable was treated as qualitative, and no class-weight correction was applied. Under these pre-pruning constraints, the CHAID algorithm determined the final tree structure.

**Reliability and cross-validation.** Model robustness and predictive stability of the CHAID trees were evaluated using ten-fold cross-validation, comparing resubstitution and cross-validated risk estimates to detect potential overfitting. The optimal tree was identified as the shallowest model, with cross-validation error within one standard error of the minimum. Resubstitution error (training misclassifications) and cross-validation error were compared. Reliable models were indicated by similar values for both errors and a quotient approaching 1.

**Regularized canonical correlation analysis (rCCA):** Regularized canonical correlation analysis (rCCA) was implemented following the framework of González et al. (2008), using the XLSTAT (Addinsoft Pearson Edition 2014) analytical engine. Regularization was employed to address multicollinearity and moderate-to-high dimensionality, improving numerical stability and limiting overfitting in covariance estimation [38]. In line with the default XLSTAT implementation, canonical variates were estimated using the complete dataset, with ridge-type shrinkage applied to the within-set covariance matrices. Shrinkage parameters ($\lambda_1$ and $\lambda_2$) were optimized internally to balance fidelity to the covariance structure and estimator stability, enabling reliable detection of structured linear associations without loss of information.

**Variable sets.** Two predictor sets were defined: (i) milk yield and composition, including unstandardized and standardized milk yield (kg) and fat, protein, lactose, dry matter (%), and SCC ($\times 10^3$ cells/mL) at 150, 210, 240, and 305 days of lactation; and (ii) fertility rates, comprising fertility per day of insemination, fertility per buck batch and day of insemination, and fertility per day by semen type (fresh/chilled vs. frozen/thawed). All fertility traits were expressed as percentages.

## Correlations and multicollinearity

Multicollinearity was assessed prior to analysis, after which Pearson's product–moment correlations were calculated both within and between sets using XLSTAT 2014, and interpreted according to standard guidelines [39,40].

**Statistical validity.** Regularization parameters were applied to mitigate overfitting and improve estimation reliability. Canonical correlations were tested using Pillai's trace, chosen for its robustness against violations of normality, heteroscedasticity, and independence [41].

**Variability explanation.** Eigenvalues, obtained from the product of the model and inverse error matrices, corresponded to squared canonical correlations, with larger values indicating greater explained variance. Canonical correlations ranged from –1–1, with values ≥0.30 considered meaningful (~10% explained variance). Redundancy coefficients quantified the variance in one variable set explained by the other.

**Roots and significance testing.** Canonical roots reflected the ordered eigenvalues, each testing the null hypothesis that the corresponding canonical correlation equaled zero. Wilks' lambda was calculated as the product of (1 − Rc) values for each dimension, where Rc denotes canonical correlation. Values near 0 indicated strong association, while values approaching 1 denoted weak association [35].

**Cross-validation.** Ten-fold cross-validation was performed in R 4.1.1 using the CCA and cOmics packages [42–45]. Regularization parameters λ1 and λ2 were optimized using the tune.rcc function to maximize cross-validation scores [46].

**Classification implications.** Classification consistency between prior and posterior fertility-based groupings was evaluated across unstandardized and standardized milk yield and composition traits, providing an additional assessment

 

of discriminant model stability. Traits analyzed included unstandardized and 150-, 210-, 240-, and 305-day standardized milk yield and composition (milk yield, protein, fat, lactose, dry matter, and SCC $\times 10^3$ cells/mL). Prior classifications were based on initial performance, whereas posterior classifications reflected the discriminant models, allowing evaluation of reclassification patterns and classification robustness.

## Ethics statement

The study was conducted in accordance with the principles of the Declaration of Helsinki and complied with Spanish legislation on animal experimentation (Royal Decree-Law 53/2013), which transposes European Union Directive 2010/63/EU of 22 September 2010. Approval was granted by the Ethics Committee for Animal Experimentation of the University of Córdoba, as the animals were used for accredited zootechnical purposes. The study was retrospective and based exclusively on routinely collected records; no experimental or management interventions were applied, and no additional animal handling beyond standard farm practices was required. Consequently, no specific ethical authorization was necessary.

## Results

### A priori assumptions

Preliminary analyses revealed deviations from normality in fertility-related traits at both the ejaculate and buck levels. Kolmogorov–Smirnov tests indicated departures from normality mainly attributable to minor tail deviations rather than systematic distributional shifts. Visual inspection of Q–Q plots suggested that the data approximated normality sufficiently to support parametric inference. Given the bounded nature of fertility traits and the inclusion of animals in genetic selection programmes based on standardized reproductive quality criteria, these distributional features were considered biologically plausible. Levene's test detected indicated heteroscedasticity ($p < 0.05$). Given the very large sample size, this result was expected and likely reflects minor scale differences rather than severe variance inequality. To mitigate potential effects on multivariate inference, robust test statistics (Pillai's trace), regularization procedures (rCCA), and cross-validation were used, and results were interpreted primarily in terms of effect size and biological coherence. Outlier detection using the ROUT method (Q = 1%) revealed no influential outliers.

### Descriptive statistics

Table 1 summarizes descriptive statistics for fertility indices, milk yield, and milk composition traits for fresh/chilled and frozen/thawed semen, together with standardized milk yield and composition at 150, 210, 240, and 305 days of lactation. Across all fertility indices, fresh/chilled semen consistently exhibited higher fertility than frozen/thawed semen. Milk yield showed marked inter-individual variability at all lactation stages, whereas milk composition traits were comparatively stable across time points. Somatic cell count displayed wide dispersion, indicating substantial heterogeneity in udder health status among animals. Overall, these descriptive patterns highlight the contrasting variability structures of yield, composition, and fertility traits that underpin subsequent multivariate analyses.

### Canonical discriminant analysis

**Multicollinearity analysis.** As shown in S2 Table, the initial multicollinearity assessment revealed extreme redundancy among milk yield and composition traits, with several variables far exceeding the conventional VIF threshold of 10. In particular, standardized fat at 305 days showed an $R^2$ of 0.994 (tolerance = 0.0056; VIF = 178.0), standardized protein at 305 days reached a VIF of 158.3, and fat at 240 days showed a VIF of 125.9; unstandardized fat also exceeded a VIF of 108. Following successive rounds of stepwise elimination, multicollinearity was markedly reduced, with retained milk variables showing VIF values below 5 (dry matter: 4.90 unstandardized, 4.79 standardized at 150 days; fat and lactose: 2.27 and 2.22, respectively).

Fertility indices exhibited lower initial collinearity than milk traits but still showed moderate redundancy (VIF = 6.83 for fertility per day of insemination and 6.07 for fertility per buck batch and day). After removal of the most redundant fertility variable, the remaining indices displayed low and balanced multicollinearity (VIF = 2.07; tolerance = 0.483).

**Table 1. Descriptive statistics (mean, standard deviation (SD), Standard error of the mean (SEM), maximum, minimum and variance ($\alpha^2$) fertility related parameters across semen type of the doses with which inseminations were performed and unstandardized and standardized milk yield and composition parameters at 150, 210, 240 and 305 days.**

| Cluster | Subcluster | | Variable | Min–Max | Mean±SD |
|---|---|---|---|---|---|
| **Fresh/chilled semen** | Fertility indexes | | Fertility per day of insemination | 0.00–93.10 | 56.17±16.52 |
| | | | Fertility per buck batch and day of insemination | 0.00–100.00 | 57.05±21.17 |
| | | | Fertility per day by semen type | 0.00–90.00 | 55.82±15.28 |
| **Frozen/thawed semen** | Fertility indexes | | Fertility per day of insemination | 0.00–82.09 | 37.75±15.96 |
| | | | Fertility per buck batch and day of insemination | 0.00–100.00 | 40.30±21.07 |
| | | | Fertility per day by semen type | 0.00–82.09 | 36.91±15.29 |
| **Milk yield & composition** | **Unstandardized** | | Milk yield (kg) | 8.70–6704.62 | 686.41±301.24 |
| | | | Fat (%) | 0.00–19.58 | 5.13±0.70 |
| | | | Protein (%) | 0.00–7.52 | 3.58±0.36 |
| | | | Lactose (%) | 0.00–9.70 | 4.75±0.31 |
| | | | Dry matter (%) | 0.00–24.45 | 14.18±1.15 |
| | | | SCC (×10³ cells/mL) | 0.00–18065.00 | 1059.97±976.56 |
| | **Standardized** | **150 days** | Milk yield (kg) | 8.70–2353.67 | 369.81±120.95 |
| | | | Fat (%) | 0.00–18.60 | 5.06±0.94 |
| | | | Protein (%) | 0.00–9.57 | 3.47±0.49 |
| | | | Lactose (%) | 0.00–10.24 | 4.84±0.54 |
| | | | Dry matter (%) | 0.00–25.55 | 14.18±1.40 |
| | | **210 days** | Milk yield (kg) | 8.70–2695.55 | 504.84±159.76 |
| | | | Fat (%) | 0.00–19.58 | 5.06±0.79 |
| | | | Protein (%) | 0.00–8.10 | 3.51±0.40 |
| | | | Lactose (%) | 0.00–9.70 | 4.79±0.38 |
| | | | Dry matter (%) | 0.00–24.45 | 14.13±1.25 |
| | | **240 days** | Milk yield (kg) | 8.70–2695.55 | 560.53±180.67 |
| | | | Fat (%) | 0.00–19.58 | 5.08±0.75 |
| | | | Protein (%) | 0.00–7.52 | 3.54±0.38 |
| | | | Lactose (%) | 0.00–9.77 | 4.77±0.35 |
| | | | Dry matter (%) | 0.00–24.45 | 14.14±1.21 |
| | | **305 days** | Milk yield (kg) | 8.70–4013.77 | 635.51±223.10 |
| | | | Fat (%) | 0.00–19.58 | 5.11±0.72 |
| | | | Protein (%) | 0.00–9.40 | 3.56±0.37 |
| | | | Lactose (%) | 0.00–9.70 | 4.75±0.33 |
| | | | Dry matter (%) | 0.00–37.00 | 14.25±1.17 |

### Canonical dimensions, efficiency and model reliability.

**Dimensions and validity:** The scree plots in Fig 5 show that the first canonical function (F1) explains most of the variance across analyses. In the rCCA, F1 captures nearly all shared variability between fertility rates and milk traits, with minimal contribution from subsequent axes. Similarly, in CDA models for fertility per day of insemination, fertility per buck batch and day, and fertility per day by semen type, F1 accounts for most discriminatory power. The sharp eigenvalue decline after F1 indicates that fertility variation is primarily structured along a single canonical dimension.

As shown in Table 2, the first canonical function (F1) had the highest eigenvalues and was highly significant across all fertility scales (Bartlett's test, $p < 0.0001$), concentrating most discriminatory information. For fertility per day of insemination, F1 reached an eigenvalue of 0.043, with subsequent functions explaining much less variance. A similar pattern was

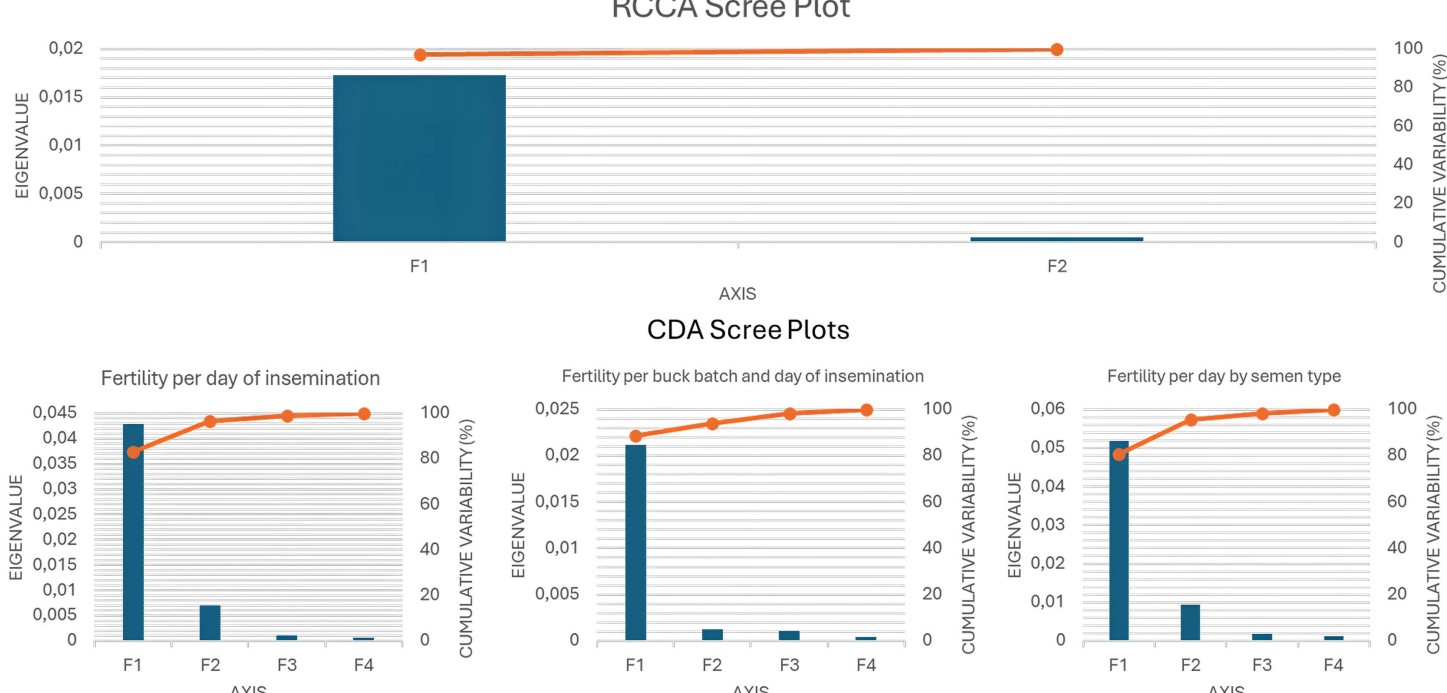

**Fig 5. Scree plots of canonical functions.** The top panel shows results from regularized canonical correlation analysis (rCCA), where fertility rates were analyzed as continuous variables together with milk yield and composition. The bottom panels show canonical discriminant analyses (CDA), where fertility rates were treated as categorical clustering variables (very low to very high) for three contexts: fertility per day of insemination, fertility per buck batch and day of insemination, and fertility per day by semen type. In all cases, the first canonical function (F1) explained the majority of variance, with higher-order functions contributing minimally.

**Table 2. Canonical discriminant analysis efficiency parameters showing the significance of each canonical discriminant function based on Bartlett's test.**

| Fertility Rate Scales | Functions | Eigenvalue | Bartlett's statistic | p-value |
|---|---|---|---|---|
| Fertility per day of insemination | F1 | 0.043 | 1651.310 | <0.0001 |
| | F2 | 0.007 | 281.723 | <0.0001 |
| | F3 | 0.001 | 55.094 | <0.0001 |
| | F4 | 0.001 | 19.684 | 0.006 |
| Fertility per buck batch and day of insemination | F1 | 0.021 | 772.302 | <0.0001 |
| | F2 | 0.001 | 88.190 | <0.0001 |
| | F3 | 0.001 | 47.460 | <0.0001 |
| | F4 | 0.000 | 13.035 | 0.071 |
| Fertility per day by semen type | F1 | 0.052 | 1842.716 | <0.0001 |
| | F2 | 0.009 | 360.406 | <0.0001 |
| | F3 | 0.002 | 85.327 | <0.0001 |
| | F4 | 0.001 | 35.451 | <0.0001 |

observed for fertility per buck batch and day (F1 = 0.021), where higher-order functions contributed minimally and F4 was not significant (p = 0.071). Fertility per day by semen type showed the strongest first axis (F1 = 0.052), with later functions providing limited additional explanatory power.

**Wilks' lambda test:** S3 Table reports tests of equality of group means across fertility scales for unstandardized milk traits, standardized 150-day traits, and semen type. For fertility per day of insemination, most unstandardized traits (milk yield, fat, protein, dry matter, SCC × 10³ cells/mL) were highly significant (p < 0.0001), while lactose showed weaker significance (p = 0.014). Standardized 150-day milk yield, lactose, and dry matter were all strongly significant (p < 0.0001), and semen type was the strongest discriminator (F = 253.820, p < 0.0001).

For fertility per buck batch and day of insemination, fewer unstandardized traits were strongly significant (milk yield, protein, SCC × 10³ cells/mL; p < 0.0001), while dry matter was not significant (p = 0.056) and lactose showed weaker effects (p = 0.001). Standardized 150-day traits remained significant, and semen type again showed strong discrimination (F = 114.794, p < 0.0001).

For fertility per day by semen type, the strongest overall differences were observed. Unstandardized milk yield (F = 58.015), protein (F = 21.541), SCC × 10³ cells/mL (F = 17.749), and dry matter (p < 0.0001) were significant, while lactose was weaker (p = 0.011). Standardized 150-day milk yield, lactose, and dry matter were all highly significant (p < 0.0001), and semen type provided the clearest separation (F = 276.109, p < 0.0001).

Overall, unstandardized milk yield, protein, and SCC × 10³ cells/mL were the most consistent discriminators, with standardized 150-day traits adding further discriminatory power. Across all fertility scales, semen type showed the strongest differentiation among fertility categories.

**Pillai's trace criterion:** The multivariate test statistics based on Pillai's trace indicate highly significant overall effects of the explanatory variables across all fertility rate scales. For fertility per day of insemination, Pillai's trace reached 0.050 (F = 41.061, p < 0.0001), confirming a strong multivariate association. For fertility per buck batch and day of insemination, the effect size was smaller (Pillai's trace = 0.023, F = 19.240, p < 0.0001), but still highly significant. The strongest overall effect was observed for fertility per day by semen type, with Pillai's trace = 0.061 (F = 45.815, p < 0.0001). In all cases, observed F values were far above the critical threshold (Fcrit = 1.394), demonstrating that the combined set of variables significantly differentiates fertility rate categories, with the semen type model showing the most pronounced multivariate discrimination.

**Canonical standardized coefficients, loadings, and spatial representation:** S4 Table reports standardized canonical coefficients linking milk traits to fertility per day of insemination, fertility per buck batch and day, and fertility per semen type across four canonical functions (F1–F4). Across all fertility measures, fresh/chilled semen showed the strongest positive coefficients in F1, indicating its dominant influence on fertility. Milk traits contributed variably: fat showed negative associations mainly in F3, protein and standardized lactose contributed positively, dry matter loaded positively in F2, and somatic cell count had minimal influence. Overall, F1 captured most shared variance, with F2–F4 reflecting secondary, trait-specific patterns.

Fig 6 compares vector loadings from rCCA and CDA. rCCA concentrated nearly all variance on F1 (97.27%), whereas CDA models explained lower but still dominant proportions on F1 (83.19%, 88.68%, and 80.78% for the three fertility traits), with F2 accounting for the remaining variance.

Discriminant loadings showed consistently larger vector magnitudes in rCCA than in CDA. Normalized rCCA vectors were 2.145 ± 0.012 for fertility per buck batch and day, 1.924 ± 0.011 for fertility per day by semen type, and 1.314 ± 0.012 for fertility per day of insemination, whereas CDA vectors remained close to unity (≈1.02 for all traits). Relative to CDA, rCCA increased loadings by ~110%, ~88%, and ~28%, respectively, indicating stronger multivariate signals and clearer trait separation.

As shown in Fig 7, Mahalanobis dendrograms revealed a consistent two-level structure across fertility measures: a Low/Very Low group and a Medium/High/Very High group. Within-group distances were small (0.02–0.10), while

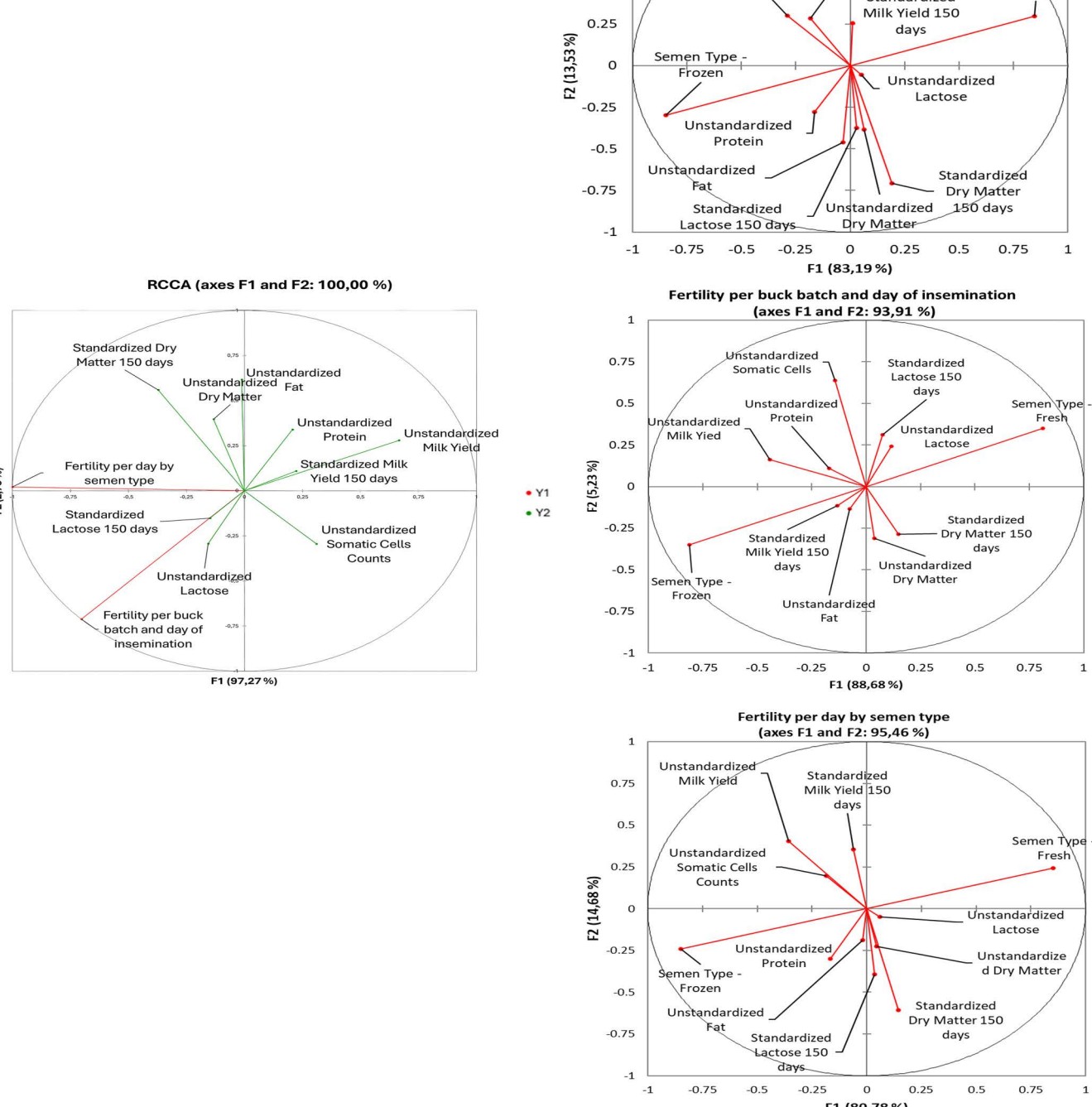

**Fig 6. Vector plot for discriminant loadings for fertility traits for regurlarized Canonical Correlation analyses (rCCA) and for Canonical Discriminant Analyses (CDA).**

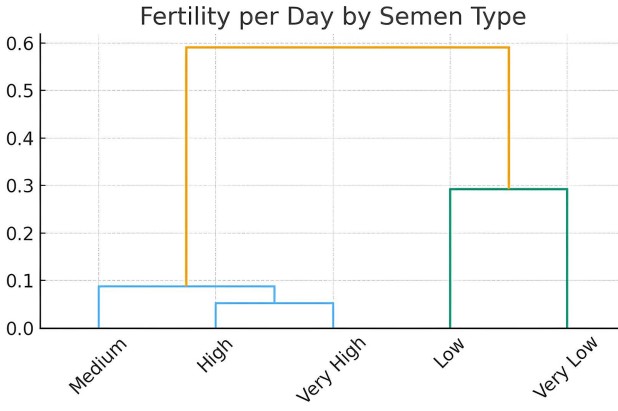

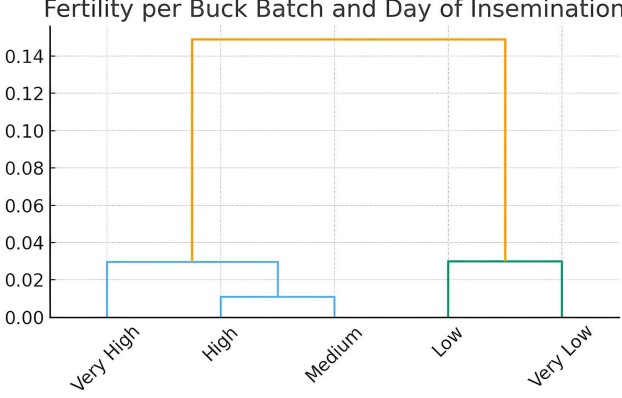

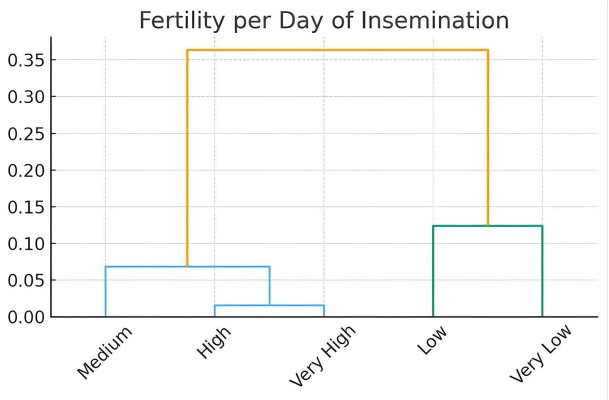

**Fig 7. Hierarchical clustering dendrograms based on Mahalanobis distances for three fertility rate measures: (A)** Fertility per Day of Insemination, **(B)** Fertility per Buck Batch and Day of Insemination, and **(C)** Fertility per Day by Semen Type. In all cases, two distinct clusters emerge, separating Very Low/Low fertility from Medium/High/Very High fertility. The degree of separation varies by measure, with the widest divergence observed for semen type (final merge at ~0.60) and the most compact structure for buck batch and insemination day (final merge at ~0.14).

separation between groups varied by fertility metric (0.35 for fertility per day of insemination, 0.14 for fertility per buck batch and day, and 0.60 for fertility per day by semen type). These results indicate that semen type provides the strongest discrimination between poor and higher fertility outcomes, whereas buck batch and insemination day yield more compact clustering.

**Discriminant function reliability: Cross-validation:**   Leave-one-out cross-validated CDA showed clear differences in classification performance across fertility traits. Press's Q values were high for all models (106.6 for fertility per day of insemination, 74.6 for fertility per buck batch and day, and 104.1 for fertility per day by semen type), indicating performance well above chance ($Q > 3.841$, $p < 0.05$). Classification accuracy exceeded random expectation (~20%), reaching 43.6%, 36.5%, and 44.8%, respectively, with the strongest performance observed for fertility per day of insemination and fertility per day by semen type.

### Data mining: CHAID decision tree

S1–S3 Figs present CHAID classification trees for fertility per day of insemination, fertility per buck batch and day of insemination, and fertility per day by semen type, illustrating hierarchical relationships between milk traits and fertility outcomes. To facilitate visualization and exploration of these trees, the SVG Tree Viewer Application [47] is provided for opening and interacting with the SVG files (Material S1).

Milk yield exhibited a non-linear association with fertility. Very low yields (8.7–394.37) were associated with reduced fertility; low–medium yields (394.37–806.74) showed moderate effects; moderate yields (806.74–2021.71) supported reproductive balance; high yields (2021.71–6704.62) were linked to reduced fertility; and extremely high yields (>6704.62) were associated with severe fertility impairment.

Dry matter at 150 days influenced fertility, with low values (0–14.325) associated with poorer outcomes, intermediate levels (14.325–25.55) optimal, and very high values (25.55–50) potentially limiting fertility. Protein showed similar patterns: low (<3.644) and very high (>6.986) values were associated with reduced fertility, whereas intermediate levels (3.644–6.986) were favorable. Fat content followed a comparable trend, with optimal fertility at medium levels (6.012–7.963), and reduced fertility at very low (0–5.258) or high (>7.963) values.

$SCC \times 10^3$ cells/mL was inversely related to fertility: very low (0–1744.783) and low (1744.783–3613.5) values were associated with better outcomes, medium values (3613.5–5845) with moderate fertility, and high values (>5845) with reduced fertility. Lactose at 150 days showed optimal fertility at intermediate levels (5.625–6.527), while very low (0–4.783) or high (>6.527) values were associated with poorer outcomes.

Very low fertility was associated with dry matter <14.325, protein <3.644, lactose <4.783, $SCC \times 10^3$ cells/mL >9453.5, milk yield <8.7, and minimal fat. Low fertility corresponded to dry matter 14.325–25.55, protein 3.644–5.4, $SCC \times 10^3$ cells/mL 1744.783–3613.5, milk yield 8.7–150, and fat 0–3.5. Medium fertility was observed with dry matter 25.55–50, protein 3.644–5.4, milk yield 150–394.37, fat 3.5–6.0, $SCC \times 10^3$ cells/mL 1744.783–3613.5, and lactose 4.783–5.625. High fertility occurred at milk yield 394.37–526, fat 6.0–6.767, protein 5.4–7.0, $SCC \times 10^3$ cells/mL 3613.5–5200, and lactose 5.625–6.0. Very high fertility was associated with dry matter >50, milk yield >500, fat >6.7, protein >7.0, and consistently higher fertility with fresh/chilled semen.

**CHAID reliability and cross-validation.**  CHAID fertility classification was evaluated using Ten-Fold cross-validation. For fertility per day of insemination, overall accuracy was 43.9%, with High fertility best predicted (79.7%), while Very High and Very Low categories showed low accuracy. Fertility per buck batch and day achieved 36.9% overall accuracy, with High fertility predicted at 87.2% but lower reliability for Medium and Low categories. Fertility per day by semen type showed 44.1% accuracy, with strongest performance for Medium (54.6%) and High (52.1%) fertility. Similar resubstitution and cross-validation risk estimates indicated stable, non-overfitted models, with lower accuracy for rare fertility categories reflecting class imbalance.

### Regularized generalized canonical correlation analysis (rCCA)

**Pearson's product–moment correlations.**  Table 3 presents Pearson's correlations between unstandardized and standardized milk yield and composition traits. Among unstandardized traits, moderate correlations were observed, particularly between fat and protein (r = 0.505) and fat and dry matter (r = 0.530), indicating that higher fat content is

Table 3. Pearson's correlations between unstandardized and standardized milk yield and composition trait pairs after VIF variable dimensionality reduction. Color scale ranges from green (maximum positive value) to red (maximum negative value).

| Variables | Unstandardized Milk Yield | Unstandardized Fat | Unstandardized Protein | Unstandardized Lactose | Unstandardized Dry Matter | Unstandardized Somatic Cells Count | Standardized Milk Yield 150 days | Standardized Lactose 150 days | Standardized Dry Matter 150 days |
|---|---|---|---|---|---|---|---|---|---|
| Unstandardized Milk Yield | 1 | −0.135 | −0.081 | −0.004 | −0.136 | −0.031 | 0.728 | 0.006 | −0.142 |
| Unstandardized Fat | | 1 | 0.505 | 0.132 | 0.530 | −0.035 | −0.228 | 0.154 | 0.465 |
| Unstandardized Protein | | | 1 | 0.201 | 0.451 | 0.036 | −0.221 | 0.169 | 0.392 |
| Unstandardized Lactose | | | | 1 | 0.149 | −0.065 | −0.017 | 0.682 | 0.163 |
| Unstandardized Dry Matter | | | | | 1 | −0.056 | −0.180 | 0.140 | 0.855 |
| Unstandardized Somatic Cells Count | | | | | | 1 | −0.023 | −0.067 | −0.072 |
| Standardized Milk Yield 150 days | | | | | | | 1 | 0.062 | −0.170 |
| Standardized Lactose 150 days | | | | | | | | 1 | 0.278 |
| Standardized Dry Matter 150 days | | | | | | | | | 1 |

associated with increased protein and dry matter. Milk yield showed weak negative correlations with fat (r = −0.135), protein (r = −0.081), and dry matter (r = −0.136), suggesting a mild dilution effect at higher production levels.

Standardized traits at 150 days, reflecting a specific lactation stage, showed strong consistency with unstandardized measures. Standardized milk yield was strongly correlated with unstandardized milk yield (r = 0.728), standardized dry matter with unstandardized dry matter (r = 0.855), and standardized lactose with unstandardized lactose (r = 0.682). These results indicate that standardization at 150 days preserves the underlying relationships among traits while facilitating comparisons across animals and lactation periods. The color scale in Table 3 visually reinforces these patterns, highlighting strong positive associations and weaker or negative correlations.

Fertility variables were strongly correlated, particularly fertility per buck batch and day of insemination with fertility per day by semen type (r = 0.686), indicating consistency across fertility measures. Table 4 shows Pearson correlations between fertility measures and unstandardized and 150-day standardized milk traits after VIF-based reduction [48]. Correlations were generally weak, with slight negative associations between milk yield and fertility per buck batch (r = −0.066 unstandardized; r = −0.022 standardized) and fertility per day by semen type (r = −0.088; r = −0.029). Other milk components (fat, protein, lactose, dry matter, SCC × 10³ cells/mL) showed minimal correlations (−0.041 to 0.049). These results indicate that, after controlling for collinearity, milk yield and composition have only minor direct associations with fertility.

Table 5 reports canonical correlation results between fertility and milk yield and composition. Two functions were extracted. The first (F1) explained 97.27% of variability and was significant (Wilks' Lambda = 0.982, F = 32.717, p < 0.0001), with a canonical correlation of 0.132, indicating a modest association. The second function (F2) contributed little (2.73%) and showed negligible correlation (0.022), despite marginal significance (p = 0.044). Redundancy coefficients were low (F1: Y1 = 0.013, Y2 = 0.002; F2: 0.000), indicating limited variance explained. Overall, the canonical relationship was statistically significant but weak, reflecting the multifactorial nature of fertility. The low canonical correlation (Rc = 0.132) and redundancy indices (1–2%) indicate that milk traits account for only a small fraction of fertility variability, consistent with

**Table 4. Pearson's correlations between fertility and unstandardized and 150 days standardized milk yield and composition parameter pairs after VIF variable dimensionality reduction. Color scale ranges from green (maximum positive value) to red (maximum negative value).**

| Variables | Fertility per buck batch and day of insemination | Fertility per day by semen type |
|---|---|---|
| Unstandardized Milk Yield | −0.066 | −0.088 |
| Unstandardized Fat | −0.009 | 0.002 |
| Unstandardized Protein | −0.024 | −0.027 |
| Unstandardized Lactose | 0.019 | 0.020 |
| Unstandardized Dry Matter | 0.006 | 0.018 |
| Unstandardized Somatic Cells Count | −0.024 | −0.041 |
| Standardized Milk Yield 150 days | −0.022 | −0.029 |
| Standardized Lactose 150 days | 0.016 | 0.019 |
| Standardized Dry Matter 150 days | 0.026 | 0.049 |

**Table 5. Canonical correlations and Redundancy coefficients.**

| Statistics | F1 | F2 |
|---|---|---|
| Eigenvalue | 0.017 | 0.000 |
| Variability (%) | 97.269 | 2.731 |
| Cumulative % | 97.269 | 100.000 |
| Wilks' Lambda | 0.982 | 1.000 |
| F | 32.717 | 1.985 |
| DF1 | 18 | 8 |
| DF2 | 65364 | 32683 |
| Pr > F | < 0.0001 | 0.044 |
| Canonical correlations: | 0.132 | 0.022 |
| Redundancy coefficients (Y1) | 0.013 | 0.000 |
| Redundancy coefficients (Y2) | 0.002 | 0.000 |

the multifactorial nature of fertility. These results suggest that milk yield and composition contribute weak but consistent signals rather than strong predictive effects on fertility.

**Canonical functions.** Two canonical functions were extracted, each representing a distinct axis linking fertility measures—fertility per day of insemination and fertility per buck batch and day of insemination—with milk composition traits, including both unstandardized and 150-day standardized parameters. The standardized canonical coefficients provide insight into the relative contribution of each variable to the canonical functions, and the following section interprets these relationships in a biologically meaningful context.

Canonical function 1 (F1) represents an *energy balance–production–fertility trade-off pathway*, contrasting higher milk yield and dry matter content with reduced fertility per buck batch and day of insemination, suggesting a negative association between productive intensity and reproductive performance. Canonical function 2 (F2) reflects an *endocrine integration and lactation-timing pathway*, opposing fertility per day versus fertility per buck batch and day of insemination and linking these patterns to variation in milk composition, particularly the distribution of dry matter across the lactation period (150-day standardized traits). Together, these functions describe distinct but complementary axes underlying the joint phenotypic structure of production and fertility. A detailed description of canonical functions F1 and F2 is provided in Material S2.

**Canonical correlation analysis ten-fold cross-validation.** Ten-fold cross-validation results for rCCA [49] are shown in Fig 8. Performance varied smoothly across the λ1–λ2 grid, with optimal values at λ1 = 0.001, largely independent of λ2, yielding a maximum score of 0.184. Agreement with Wilks' lambda for F1 (Table 5) supports model robustness. Dimensionality selection followed González et al. (2012), based on the clear separation between the first two canonical correlations, with 84.24% of variance explained by F1 (52.79%) and F2 (31.46%). Final λ values were selected following De Cecco et al. (2017) to ensure stable regularization.

## Integrated pathways analysis

Across rCCA, CDA, eigenvalue inspection, clustering, and cross-validation analyses (Figs. 5–7), fertility–production relationships consistently exhibited a low-dimensional and stable structure. Most shared covariance and discriminatory power were concentrated along the first canonical function, with a secondary orthogonal component refining this pattern. Clustering and discrimination results aligned with this canonical architecture, indicating that fertility classes separate along a dominant biological gradient with context-dependent modulation rather than independent effects [50]. These convergent statistical results provide the basis for interpreting fertility variation in terms of three interacting biological pathways: a primary energy balance and metabolic partitioning pathway (Pathway I), a secondary endocrine integration and lactation-timing pathway (Pathway II), and an autonomous semen-related pathway that modulates fertility outcomes without altering the underlying canonical structure (Pathway III).

### Pathway I: Energy balance and metabolic partitioning.

*(Dominantly expressed along Canonical Function 1)*

Canonical Function 1 (F1) accounted for the vast majority of shared covariance between production and fertility traits and represents the dominant biological axis structuring the data [31,34]. The clear dominance of F1 across all scree plots (Fig 5), together with its consistently high explanatory and discriminatory power in both rCCA and CDA models (Fig 6), demonstrates that fertility variation is primarily organized along a single canonical dimension rather than distributed across multiple independent axes.

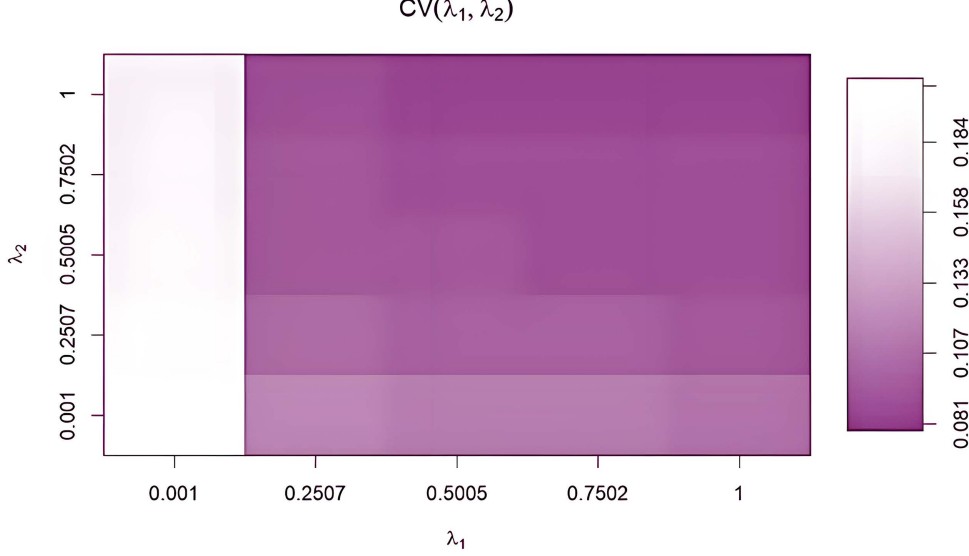

**Fig 8. Optimal cross-validation scores for the values of the parameter of regularization (λ₁ and λ₂).**

On the fertility side, F1 was almost entirely dominated by fertility per buck batch and day of insemination, with a strong negative loading. This indicates that F1 primarily contrasts high versus low fertility across insemination batches and days, capturing systematic variation in reproductive success rather than random or day-specific noise. The population-level nature of this fertility gradient is further supported by the Mahalanobis dendrograms (Fig 7), which consistently separate Low/Very Low from Medium/High/Very High fertility classes along a single dominant branch.

On the milk production side, milk yield and dry matter percentage loaded strongly and positively, whereas standardized dry matter at 150 days and lactose exhibited strong negative weights. This configuration defines a gradient in which higher milk output and solids concentration in early or mid-lactation are associated with reduced fertility across insemination contexts. The concordance of these loadings across rCCA and CDA (Fig 6) indicates that the same production traits underpin fertility discrimination irrespective of the modeling framework.

Biologically, F1 represents an energy balance and metabolic partitioning pathway, capturing the trade-off between milk synthesis and reproductive performance. Goats positioned at the positive end of this axis prioritize nutrient allocation toward milk production, whereas those at the negative end exhibit comparatively higher fertility. The opposing signs of contemporaneous versus standardized lactation traits further suggest that temporal redistribution of energy across lactation plays a critical role in shaping reproductive competence. The two-level fertility structure observed in Fig 7 reflects coordinated physiological allocation rather than discrete fertility phenotypes. Overall, this pathway reflects a dominant associative metabolic constraint rather than direct causal effects of individual milk traits on fertility [51,52].

**Pathway II: Endocrine integration of lactation and reproduction.**

*(Primarily expressed along Canonical Function 2)*

Canonical Function 2 (F2), while explaining a substantially smaller proportion of variance, refines biological interpretation by capturing structured variation orthogonal to the dominant production–fertility trade-off defined by F1. The much lower eigenvalues associated with F2 across all analyses (Fig 5), together with its secondary contribution in CDA models (Fig 6), confirm that F2 does not define an alternative fertility axis but rather modulates fertility expression within the constraints imposed by energy balance.

On the fertility side, F2 contrasts fertility per day of insemination negatively with fertility per buck batch positively, distinguishing short-term insemination-day effects from broader, more stable batch-level fertility patterns. On the milk composition side, F2 is driven primarily by standardized dry matter at 150 days with a strong positive loading, while contemporaneous dry matter percentage loads negatively.

F2 therefore represents an endocrine integration and lactation-timing pathway, linking fertility expression to temporal changes in milk composition across the lactation curve. Animals scoring highly along this axis maintain favorable milk solids profiles later in lactation while exhibiting fertility patterns that are less sensitive to daily insemination timing. This refined modulation of fertility is consistent with coordinated regulation of prolactin, growth hormone, and ovarian activity across lactation [53–55]. The limited but reproducible contribution of F2 across models (Figs 4 and 5) underscores its role as a secondary, integrative pathway rather than a primary determinant of fertility differences.

**Pathway III: Semen-related and management-mediated fertility modulation.**

*(Expressed through discrimination strength rather than canonical dominance)*

Against the background of integrated female physiology captured by F1 and F2, semen-related variables emerged as a clearly differentiated source of fertility variation. Semen type exerted a dominant influence on reproductive success, with fresh/chilled semen consistently outperforming frozen/thawed semen across fertility indices [56]. This effect is particularly evident in the Mahalanobis dendrograms (Fig 7), where fertility per day by semen type shows the strongest separation between low and higher fertility groups, and in the high Wilks' lambda F values associated with semen-related discrimination.

The stability of this pattern is further supported by cross-validation results for rCCA (Fig 8), which demonstrate that fertility discrimination linked to semen-related effects is robust across regularization settings and not driven by overfitting. The independence of this pathway aligns with extensive evidence that cryopreservation impairs sperm motility, membrane integrity, and fertilizing capacity irrespective of female metabolic or endocrine status [20,57].

Taken together, these results support the interpretation of semen quality as an autonomous fertility pathway that interacts with, but is not regulated by, female production physiology. Semen-related factors therefore act as external modulators of fertility expression, amplifying or constraining reproductive outcomes along the canonical metabolic and endocrine axes defined by F1 and F2.

## Discussion

The present study demonstrates that the relationships between milk yield, milk composition, and fertility in Murciano-Granadina dairy goats are structured around a limited number of recurrent and biologically coherent pathways rather than independent trait effects. Across multicollinearity diagnostics, canonical multivariate analyses, classification trees, and validation procedures, three consistent pathways emerged: (i) an energy balance and metabolic partitioning pathway, (ii) an endocrine integration pathway linking lactation dynamics with ovarian function, and (iii) a semen-related fertility pathway that operates largely independently of female production physiology. The dominant canonical axes therefore represent coordinated physiological regulation rather than additive effects of individual traits [58–60].

Multivariate associations identified were statistically significant and consistent across analytical approaches, even if their individual effect sizes were moderate as it could have been expected. Canonical correlations were low (Rc = 0.132), redundancy indices did not exceed 2%, and pairwise Pearson correlations between milk traits and fertility ranged from 0.02 to 0.08. These results reflect the multivariate nature of fertility, which arises from the combined contribution of multiple physiological and management factors. In this context, the coefficients of variation obtained from the three cross-validation procedures demonstrate that multivariate milk yield and composition profiles support accurate individual-level fertility prediction when traits are evaluated jointly rather than in isolation.

### Multivariate structure of production and fertility traits

Milk yield and milk composition variables clustered into a strongly collinear multivariate block, reflecting their shared physiological, metabolic, and endocrine regulation of mammary function and nutrient allocation. This structural pattern indicates that milk yield, fat, protein, lactose, dry matter, and their standardized counterparts represent overlapping dimensions of a common underlying biological process rather than independent sources of information [61,62]. Comparable collinearity structures have been reported in dairy goats and cattle, where milk production and composition jointly capture variation in metabolic status and hormonal control rather than discrete functional traits [63,64].

In contrast, fertility-related variables displayed a more fragmented multivariate structure, with limited redundancy largely confined to insemination-timing descriptors. Fertility outcomes stratified by semen type remained largely orthogonal to production traits, supporting their interpretation as a distinct reproductive dimension [65,66]. Hierarchical clustering and CHAID analyses consistently partitioned fertility into low versus moderate-to-high outcome classes, with relatively weak separation among intermediate categories. This structural pattern suggests the presence of threshold-based physiological constraints on fertility, modulated by semen-related factors, rather than a continuous linear response to variation in production intensity [64].

### Biological interpretation of integrated pathways analysis

**Pathway I: Energy balance and metabolic partitioning.** The dominance of F1 strongly echoes the central role of energy balance in shaping reproductive outcomes in dairy species. Numerous studies have shown that high milk

production early in lactation imposes a metabolic load that suppresses ovarian cyclicity, delays resumption of estrus, and reduces conception rates [67,68]. The negative association between fertility and high-yield traits observed here fits this pattern precisely. The strong positive loadings of milk yield and dry matter percentage on F1 reflect animals that prioritize nutrient allocation toward lactation, consistent with the metabolic partitioning framework described by Bauman and Currie [69] and further elaborated in modern dairy physiology [70,71].

The contrasting signs of contemporaneous versus standardized mid-lactation traits suggest that the timing of nutrient use is as important as the magnitude. This aligns with evidence that cows and goats with more persistent lactation curves or delayed peak yield often experience deeper or more prolonged negative energy balance, which compromises reproductive performance [72,73]. The clear two-level fertility structure in the dendrograms supports the idea that fertility is constrained by a coordinated physiological strategy rather than isolated trait effects.

Overall, the pattern observed in F1 is highly consistent with the broader literature showing that fertility is fundamentally limited by metabolic load and nutrient partitioning, not by any single milk component [59,60,74].

**Pathway II: Endocrine integration of lactation and reproduction.** F2 captures a subtler but biologically coherent layer of variation that reflects endocrine coordination across lactation. The differentiation between insemination day fertility and batch-level fertility resonates with studies showing that reproductive success is influenced not only by metabolic status but also by hormonal rhythms and lactation stage [75,76]. The contrast between standardized dry matter at 150 days and contemporaneous dry matter percentage suggests that animals differ in how their milk composition evolves across lactation, a trait often linked to endocrine regulation of persistency and mammary metabolism [77,78].

The endocrine interpretation of F2 is supported by extensive evidence that prolactin, growth hormone, IGF-1, and gonadotropins jointly regulate both lactation and ovarian function [79–81]. Animals that maintain favorable solids profiles later in lactation may also maintain more stable endocrine environments, reducing sensitivity to insemination day fluctuations. This aligns with findings that reproductive success is influenced by the interaction between lactation stage, hormonal milieu, and follicular dynamics [82,83].

Thus, F2 appears to represent an integrative endocrine pathway that modulates fertility within the broader metabolic constraints defined by F1. Its smaller but consistent contribution across models mirrors the literature showing that endocrine coordination fine-tunes reproductive outcomes but rarely overrides metabolic limitations.

**Pathway III: Semen-related and management-mediated fertility modulation.** The strong discriminant power of semen type observed in this study is consistent with extensive evidence demonstrating the negative impact of cryopreservation on sperm function. Frozen/thawed semen is known to exhibit reduced motility, altered membrane and acrosomal integrity, increased oxidative stress, and diminished fertilizing capacity when compared with fresh or chilled semen [84,85]. These cryo-induced alterations have been repeatedly documented in goats [65,66] and across other domestic ruminant species [86,87].

However, the magnitude of the semen-type effect observed here cannot be attributed solely to intrinsic biological damage caused by freezing and thawing. In practical artificial insemination programs, frozen semen performance is also highly sensitive to management-related factors, including storage conditions, transport logistics, thawing temperature and duration, and operator handling at insemination. Deviations from recommended protocols can exacerbate cryoinjury and further reduce sperm viability and functional competence [88,89].

The clear separation of fertility groups by semen type in the dendrograms, together with the strong Wilks' lambda values, reflects the cumulative effect of both biological susceptibility to cryopreservation and variability introduced by semen handling practices. Notably, semen-related variation operated largely independently of the physiological axes represented by F1 and F2, suggesting that semen quality constitutes an external fertility pathway not directly governed by female metabolic or endocrine status [90,91].

Taken together, these results support the interpretation of semen type as a management-mediated determinant of fertility. While fresh and chilled semen are less vulnerable to handling-induced losses, frozen semen introduces additional layers of technical sensitivity that can amplify fertility variability under field conditions [88].

**Non-linear associations between milk yield and fertility.** Beyond linear associations, fertility displayed pronounced non-linear relationships with milk yield. Intermediate production levels were generally associated with higher fertility, whereas both very low and very high yields clustered with poorer reproductive outcomes, consistent with energy balance theory [59]. Linear correlations remained weak, indicating that these relationships are poorly captured by bivariate approaches [31].

Goats producing more than approximately 500 L per lactation showed comparatively favorable fertility, despite the well-documented antagonism between high yield and reproduction in dairy species [92,93]. This apparent contradiction likely reflects management-mediated effects, including targeted nutritional supplementation, while excessive energy intake may impair oocyte and embryo quality [74]. Fertility in high-yielding animals therefore appears to depend on feed efficiency and nutrient partitioning rather than milk output per se [10,94].

**Milk composition, udder health, and fertility.** Milk composition traits contributed to fertility discrimination primarily through their combined profiles rather than as independent predictors. Optimal fertility was associated with intermediate protein, fat, lactose, and dry matter levels, consistent with observations in dairy cattle and goats [95–97]. Lactose emerged as a sensitive indicator of energy balance, with low values reflecting negative energy status and impaired reproductive hormone regulation [98–100].

Somatic cell count exerted a consistent but modest influence on fertility, reflecting an inflammatory–immune pathway that indirectly constrains reproductive performance [101,102]. Elevated SCC × 10³ cells/mL values, indicative of subclinical mastitis, were associated with reduced fertility and remain a major constraint in dairy goat production [103,104].

**Genetic and endocrine integration hypotheses.** The phenotypic pathways identified in this study are consistent with, but do not demonstrate, underlying genetic and endocrine mechanisms linking production and reproduction. Polymorphisms in genes involved in energy balance and endocrine signaling, including IGF1 (*insulin-like growth factor 1*), LEP (*leptin*) and its receptor LEPR (*leptin receptor*), as well as FSHR (*follicle-stimulating hormone receptor*) and LHR (*luteinizing hormone receptor*), have been reported in previous studies to be associated with both milk production traits and fertility outcomes and therefore cannot be excluded as contributors to the multivariate phenotypic patterns observed here [14–16,105]. Likewise, additional candidate genes related to lipid metabolism and hormonal regulation—such as DGAT1 (*diacylglycerol O-acyltransferase 1*), SCD (*stearoyl-CoA desaturase*), GHR (*growth hormone receptor*), and PRL (*prolactin*)—may influence these relationships indirectly, supporting the biological plausibility of the dominant multivariate axes without implying direct genetic or causal effects [19,106].

## Implications for breeding and management

Artificial insemination remains a cornerstone of genetic improvement in dairy goats, and the present results highlight its potential to deliver high fertility when semen quality and female physiological balance are jointly considered [26,28]. The clear discrimination among semen types underscores the importance of continued investment in semen processing and evaluation, while also emphasizing selection of females capable of sustaining reproductive efficiency under high production demands [5,107].

Nutritional and management practices play a complementary enabling role by allowing genetically superior animals to fully express favorable pathway-level traits. Coordinated implementation of breeding, nutrition, and reproductive management strategies therefore provides a strong basis for sustainable improvement in productivity, milk quality, and fertility [108,109].

## Limitations and future research directions

A substantial proportion of fertility variance reflects its multifactorial nature, arising from the combined influence of genetic background, seasonality, nutritional management, health status, and insemination logistics acting alongside milk traits to shape reproductive performance. This retrospective observational study characterizes integrative biological pathways from multivariate phenotypic structure rather than direct measurement of molecular or physiological

mechanisms. The large sample size enables robust population-level inference consistent with the polygenic architecture of fertility. Future studies integrating longitudinal phenotypic data with metabolic, endocrine, and genomic information will further refine pathway-level inference and enhance translation into targeted breeding, nutritional, and reproductive management strategies.

## Conclusions

This study shows that fertility in Murciano-Granadina dairy goats is structured around a limited number of recurrent multivariate pathways linking milk yield, milk composition, and reproductive outcomes, rather than independent effects of individual traits. The dominant pathway reflects variation associated with energy balance and metabolic partitioning, in which extreme milk production levels and imbalanced milk solids are associated with reduced fertility, while intermediate production levels correspond to more favorable reproductive performance. Additional pathways capture temporal variation in milk composition across lactation and the influence of semen type, indicating that fertility expression arises from the combined effects of female physiological status and management-related factors. Overall, associations between milk traits and fertility were statistically robust but of modest magnitude, consistent with the multifactorial nature of reproductive performance. These findings provide a population-level description of how fertility co-varies with production traits in Murciano-Granadina goats and underscore the importance of considering production and reproduction jointly when evaluating fertility patterns.

## Supporting information

**S1 Material.** *SVG Tree Viewer Application* **for visualization and interactive exploration of CHAID classification trees in SVG format.** The software facilitates inspection of hierarchical node structure, split criteria, and classification paths in S1–S3 Figs (Navas González, 2023).
(RAR)

**S2 Material. Canonical functions F1 and F2 describing multivariate associations between fertility indices and milk production and composition traits.**
(DOCX)

**S1 Fig. CHAID classification tree showing the segmentation of fertility per day of insemination.** The tree identifies day-specific thresholds that define subgroups with homogeneous fertility outcomes, highlighting temporal patterns associated with reproductive performance.
(SVG)

**S2 Fig. CHAID classification tree illustrating fertility per buck batch and day of insemination.** The hierarchical structure reveals interactions between buck batch and insemination day, leading to distinct fertility profiles across successive nodes.
(SVG)

**S3 Fig. CHAID classification tree describing fertility per day of insemination according to semen type.** The resulting splits indicate differential fertility responses across days depending on semen preservation method, defining semen-specific fertility patterns.
(SVG)

**S1 Table. Farm geolocalization where Murciano-Granadina does were located across Spain and Portugal territories.**
(XLSX)

**S2 Table. Multicollinearity Analysis of Milk Yield and Composition related Parameters and Fertility Scale and/or rates Sets.**
(XLSX)

**S3 Table. Results for the tests of equality of group means to test for difference in the means across fertility rate scale levels once redundant variables have been removed.**
(DOCX)

**S4 Table. Standardized canonical coefficients for the relationship between milk production traits (milk yield, fat, protein, lactose, dry matter, somatic cell count, and standardized 150-day measures) and fertility (per day of insemination, per buck batch/day, and per semen type) across four canonical functions (F1–F4).** Positive and negative values indicate the relative contribution of each trait to the canonical variable, with fresh semen consistently dominating F1.
(DOCX)

## Acknowledgments

The authors would like to acknowledge to the National Association of Breeders of Murciano-Granadina Goat Breed and to the Andalusian goat selection and improvement center (Fuente Vaqueros, Granada, Spain).

## Author contributions

**Conceptualization:** Juan Vicente Delgado Bermejo, Francisco Javier Navas González.

**Data curation:** María Pía Peláez Caro, Ander Arando Arbulu, José Manuel León Jurado, Francisco Javier Navas González.

**Formal analysis:** María Pía Peláez Caro, Ander Arando Arbulu, José Manuel León Jurado, Francisco Javier Navas González.

**Funding acquisition:** Juan Vicente Delgado Bermejo, Javier Fernández Álvarez.

**Investigation:** María Pía Peláez Caro, Ander Arando Arbulu, José Manuel León Jurado, Javier Fernández Álvarez, Francisco Javier Navas González.

**Methodology:** María Pía Peláez Caro, Ander Arando Arbulu, José Manuel León Jurado, Francisco Javier Navas González.

**Project administration:** Juan Vicente Delgado Bermejo, Javier Fernández Álvarez.

**Resources:** Ander Arando Arbulu, Juan Vicente Delgado Bermejo, Javier Fernández Álvarez.

**Software:** María Pía Peláez Caro, Ander Arando Arbulu, José Manuel León Jurado, Juan Vicente Delgado Bermejo, Francisco Javier Navas González.

**Supervision:** Ander Arando Arbulu, José Manuel León Jurado, Juan Vicente Delgado Bermejo, Javier Fernández Álvarez, Francisco Javier Navas González.

**Validation:** Ander Arando Arbulu, Juan Vicente Delgado Bermejo, Javier Fernández Álvarez, Francisco Javier Navas González.

**Visualization:** Ander Arando Arbulu, José Manuel León Jurado, Juan Vicente Delgado Bermejo, Javier Fernández Álvarez, Francisco Javier Navas González.

**Writing – original draft:** María Pía Peláez Caro, Francisco Javier Navas González.

**Writing – review & editing:** Ander Arando Arbulu, José Manuel León Jurado, Juan Vicente Delgado Bermejo, Javier Fernández Álvarez, Francisco Javier Navas González.

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
