## [Decision Letter · Decision Letter 0]

25 Dec 2025

Dear Dr. Navas González,

Thank you for submitting your manuscript to PLOS ONE. After careful consideration, we feel that it has merit but does not fully meet PLOS ONE’s publication criteria as it currently stands. Therefore, we invite you to submit a revised version of the manuscript that addresses the points raised during the review process.

We look forward to receiving your revised manuscript.

Kind regards,

Ghadeer Sabah Bustani, Ph.D

Academic Editor

PLOS One

Journal Requirements:

Funding was not received for the development of the present study. The present research was carried out during the covering period of a Ramón y Cajal Post-Doctoral Contract with the reference MCIN/AEI/10.13039/501100011033 and the European Union “NextGenerationEU”/PRTR.

4. In the online submission form, you indicated that data will be made available from the corresponding author F.J.N.G. upon reasonable request.

6. We note that Figure 2 in your submission contain map images which may be copyrighted. All PLOS content is published under the Creative Commons Attribution License (CC BY 4.0), which means that the manuscript, images, and Supporting Information files will be freely available online, and any third party is permitted to access, download, copy, distribute, and use these materials in any way, even commercially, with proper attribution. For these reasons, we cannot publish previously copyrighted maps or satellite images created using proprietary data, such as Google software (Google Maps, Street View, and Earth). For more information, see our copyright guidelines: http://journals.plos.org/plosone/s/licenses-and-copyright.

Reviewers' comments:

Reviewer's Responses to Questions

**Comments to the Author**

1. Is the manuscript technically sound, and do the data support the conclusions?

Reviewer #1: Yes

Reviewer #2: Yes

Reviewer #3: Yes

Reviewer #4: Yes

Reviewer #5: Yes

Reviewer #6: Yes

Reviewer #7: Partly

2. Has the statistical analysis been performed appropriately and rigorously?

Reviewer #1: Yes

Reviewer #2: Yes

Reviewer #3: Yes

Reviewer #4: Yes

Reviewer #5: Yes

Reviewer #6: Yes

Reviewer #7: Yes

3. Have the authors made all data underlying the findings in their manuscript fully available?

Reviewer #1: Yes

Reviewer #2: No

Reviewer #3: Yes

Reviewer #4: Yes

Reviewer #5: Yes

Reviewer #6: Yes

Reviewer #7: Yes

4. Is the manuscript presented in an intelligible fashion and written in standard English?

Reviewer #1: Yes

Reviewer #2: Yes

Reviewer #3: Yes

Reviewer #4: Yes

Reviewer #5: Yes

Reviewer #6: Yes

Reviewer #7: No

Reviewer #1: Abstract: Well written and adequate

Introduction: Mostly well written. There is a paragraph on genetic influence and list of genes associated with milk production and fertility. These genes were not investigated in the current study. Suggest to trim down the genetic mention.

Materials & methods:

Extensive, well-thought out & well planned methodologies.

These were conducted well and written/explained well in the methodology section.

Measures have been taken to remove/reduce bias.

Results:

There are extensive explanation of the results which is not needed in the result section and should be better saved for the discussion.

All figures given were of high quality.

Discussion:

The discussion is adequate. There is a tendency to mention genetic and hormonal components that play important roles in fertility and milk production at many instances, but these were not really investigated in the study. Instead, they highlighted what the study is lacking. Therefore a clear mention of study limitations and future direction would be good to supplement these “uninvestigated” aspects of milk prodution vs fertility.

Reviewer #2: . 1- Strengths (Positive Points)

A. Strong and Highly Relevant Research Topic

- The manuscript addresses a significant challenge in dairy small ruminants: the balance between milk production and fertility, particularly in Murciano-Granadina goats, an economically important breed.

- The focus on canonical pathways linking production and fertility is innovative and aligns with current priorities in animal breeding and reproductive physiology.

B. Extremely Large and Valuable Dataset

- The study includes 32,693 insemination records and 29,390 milk composition records, covering over 21,000 goats.

- This exceptional sample size strengthens the generalizability and statistical power of the results.

C. Advanced Statistical and Multivariate Methods

- The authors employ a combination of Canonical Discriminant Analysis (CDA), Regularized Canonical Correlation Analysis (RCCA), CHAID decision trees, and multicollinearity filtering using VIF.

- This multilevel analytical framework is sophisticated and appropriate for revealing structure in complex biological relationships.

D. Clear Identification of Physiological and Endocrine Mechanisms

- The manuscript provides a comprehensive background on:

- Endocrine pathways (IGF-1, leptin, HPG axis).

- Genetic polymorphisms (IGF1, LEP, DGAT1, etc.).

- Physiological balance between production and reproduction.

- This supports the biological interpretation of the statistical findings.

E. Strong Practical Implications

- The study identifies specific thresholds for milk traits associated with optimal fertility (e.g., protein 3.64–6.98%, fat 6.0–7.9%, SCC < 5200×10³).

- Results have direct value for breeding programs, AI centers, and dairy goat farmers.

F. Data Quality Control

The authors conduct:

Outlier screening (ROUT),

Normality checks (Kolmogorov–Smirnov),

Homoscedasticity testing (Levene),

Multicollinearity evaluation (VIF),

Cross-validation.

These steps confirm good methodological rigor.

2. Weaknesses (Negative Points)

A. Manuscript Organization and Length

- The manuscript is unusually long, especially the Materials and Methods section, which contains:

- Excessive detail on AI protocols,

- Geography of farms,

- Temperature and climatic descriptions,

- Very long mathematical formulas.

- Some methodological details could be moved to Supplementary Materials to improve readability.

B. Redundancy in Methods and Results

- Some concepts are repeated multiple times (e.g., fertility definitions, multicollinearity screening, standardized vs. unstandardized traits).

- Several paragraphs in Results re-explain information already stated in Methods.

C. Limited Clarity in Figures and Data Interpretation

- Some referenced figures (e.g., Figures 1, 2, 3, 4, 5) are not fully visible or not well explained.

- Vector loading interpretations would benefit from shorter, clearer explanations.

- Complex statistical outputs are sometimes described narratively without graphical support.

D. Canonical Functions Are Not Fully Interpreted Biologically

- Although the statistical results are well described numerically, the manuscript could improve by relating each canonical dimension back to biological meaning.

- Example: What biological mechanism is represented by F1? By F2?

E. Discussion Section Needs Sharper Structure

- Currently, the discussion merges:

General literature review,

Physiological mechanisms,

Interpretation of findings,

Genetic discussion.

It would benefit from reorganizing into clear subsections:

Main findings,

Comparison with previous studies,

Biological interpretation,

Implications for breeding,

Limitations,

Future research.

F. English Language and Style Issues

- Several sentences are lengthy, complex, or contain grammatical issues.

- The manuscript would benefit from professional language editing to improve academic flow.

G. Lack of Clear Hypothesis Statements

The introduction presents background and aims, but the central hypothesis is not explicitly stated.

Overall Recommendation: Major Revision

Reasons for Major Revision

Although the science is strong and the dataset is excellent, the manuscript requires:

- Reorganization for clarity and conciseness.

- Improved figure presentation and interpretation.

- Stronger linkage between statistical results and biological meaning.

- Significant English-language editing.

- Removal or relocation of redundant methodological details.

4. Specific Suggestions to Improve the Manuscript

A. Introduction

Add a clear, concise hypothesis.

Shorten literature review on genetic markers.

B. Methods

Move AI protocol details, climate descriptions, and long equations to supplementary materials.

Keep Methods focused on:

Data description,

Key variables,

Statistical workflow.

C. Results

Use more graphical summaries.

Reduce repeated text explaining the same findings in multiple tables.

D. Discussion

Rewrite into structured subsections.

Focus on interpretation, not re-stating numerical results.

E. English Editing

Improve flow, reduce sentence length, and correct grammar.

F. Figures

Ensure readability, proper labels, and brief explanatory captions.

Reviewer #3: This manuscript is highly interesting and scientifically valuable. It provides a clear explanation about the associations between milk yield, milk composition, and fertility in Murciano-Granadina goats, evaluated through canonical discriminant analysis (CDA) and regularized canonical correlation analysis (RCCA). The findings convincingly emphasize that milk traits are essential physiological indicators of reproductive capacity.

Importantly, the study demonstrates that incorporating the milk quality traits into genetic selection programs can support the simultaneous optimization of productive performance and reproductive efficiency. Regarding the quality of the figures (Figure 3 and Figure 4, Page 70 and 71), I recommend improving their clarity, as they are currently difficult to read.

Thank you very much.

Best regards,

Reviewer #4: 1. General Comments

The study is robust and relevant, offering strong statistical analysis of milk and fertility traits in Murciano-Granadina goats. However, the manuscript is overly long, includes redundant methods, and needs clearer structure.

2. Major Comments

1. Introduction too long — Includes unnecessary hormonal, metabolic, and genetic background; needs sharper focus.

2. Methods overly detailed — CDA/CCA/RCCA theory is repeated unnecessarily; should be shortened.

3. Descriptive table too large — Table I is difficult to read; parts should move to Supplementary Materials.

4. Weak biological interpretation — Lactose, SCC, and dry matter need clearer physiological explanations.

5. Lactation stages not justified — The choice of 150, 210, 240, and 305 days needs biological or practical justification.

6. Insufficient goat-specific comparisons — Discussion relies mainly on bovine studies; more goat literature is needed.

3. Minor Comments

1. Sentences too long; reduce repetition and passive voice.

2. Figures 3–5 need clearer biological explanations; axis labels should be larger.

3. Update older (pre-2010) references and add more goat-related citations.

4. Standardize units (e.g., SCC × 10³ cells/mL).

5. Separate fresh vs. frozen semen results.

Reviewer #5: The statistical analysis has been conducted sophisticatedly and rigorously, and the data supports the conclusion. However, the manuscript needs a major revision to improve its clarity, focus, and defensibility. The supplementary figures, such as S1, S2, and S3, do not show the CHAID classification trees as mentioned in the manuscript. The authors are requested to upload the original supplementary figure files. I have also attached a file of my review comments. Please look for my comments.

Reviewer #6: The current manuscript highlights the impact of milk production on fertility of does in a scientifically sound, intelligible and rigorous pattern.

Line 94 & 95: Abbreviations (DGAT1, SCD, GHR, PRL) should be introduced correctly

Reviewer #7: Abstract

The abstract describes a potentially valuable and comprehensive study; however, major improvements are required.

1. While the abstract states that the study “investigates intricate connections,” it does not clearly define the primary biological question or hypothesis. It remains unclear whether the analysis is exploratory or explicitly designed to test a production–fertility trade-off. A concise statement emphasizing the scientific rationale, urgency, and main objective of the study is needed.

2. Although the reported Wilks’ Lambda value (0.982) is statistically significant—likely due to the very large sample size—it indicates limited discriminant power, which should be interpreted more cautiously.

3. The abstract is numerically dense, reducing readability and obscuring the main biological message; reducing detailed statistics and emphasizing key biological patterns is recommended.

4. Minor formatting and typographical inconsistencies (e.g., decimal notation) should be corrected.

Introduction:

The Introduction establishes the relevance of studying milk–fertility relationships in Murciano-Granadina goats and is supported by an extensive literature base. However, it would be substantially strengthened by improving these points:

1. The Introduction provides a broad overview of genetic selection, milk production, and fertility, but the specific research gap addressed by this study is not clearly defined until late in the section. The narrative would benefit from an earlier and more explicit statement of the unresolved question the study aims to address.

2. Detailed descriptions of endocrine and genetic pathways are presented; however, these mechanisms are not directly measured in the study. The Introduction should clearly distinguish between data-supported findings and theoretical or interpretative frameworks to avoid overstating biological inference.

3. Several statements imply causal relationships between milk traits and fertility, which cannot be established using observational canonical analyses. The language should be revised to emphasize associations rather than mechanistic causation.

4. While CDA and CCA are introduced, the biological rationale for choosing these methods over more conventional multivariate or mixed-model approaches is not sufficiently explained and should be clarified.

Materials and Methods

The methodological framework is appropriate for the study objectives. However, several aspects require clarification, simplification. This section is overly long and highly fragmented, with excessive descriptive detail that does not clearly contribute to the statistical analyses. Streamline non-essential environmental descriptions and improve structural coherence by clearly separating biological procedures from statistical methodology.

Major Comments:

1. Over-complexity and analytical overload: The study applies an extensive combination of multivariate, discriminant, clustering, and data-mining techniques (CDA, CCA, RCCA, PCA, CHAID, UPGMA), but the overall analytical framework is not clearly structured. The manuscript does not sufficiently explain why all these methods are required, how they complement each other, or how redundancy between approaches is avoided.

2. Insufficient justification for categorization of fertility traits: Continuous fertility measures are converted into ordinal categories without clear biological or statistical justification. This transformation may reduce information content and conflicts with the assumptions of canonical and correlation-based analyses.

3. Statistical significance versus biological relevance

Several statistical tests (Wilks’ lambda, Pillai’s trace, Bartlett’s test) emphasize significance thresholds, but limited attention is given to effect sizes, discriminant strength, or practical relevance, particularly when lambda values indicate weak group separation.

4. Biological interpretability of classification outcomes

Reclassification of animals into higher or lower fertility categories and decision-tree outputs are statistically evaluated, but their implications for reproductive management, breeding decisions, or genetic selection are not clearly articulated.

Minor Comments – Materials and Methods

1. Terms such as “clustering variables,” “fertility scales,” and “classification reliability” are sometimes used inconsistently and should be more clearly defined.

2. Redundancy in methodological descriptions: Multicollinearity assessment, correlation thresholds, and validation procedures are described multiple times across sections and could be consolidated.

3. There are grammatical inconsistencies, spacing errors, and occasional unclear phrasing.

Results:

1. The Results are overly detailed, with substantial repetition between the narrative, tables, and figures. Many numerical values and statistical outcomes are restated verbatim across text, tables, and supplementary materials The Results should be significantly shortened and streamlined, focusing on the key outcomes, while secondary details can be relegated to tables or supplementary files.

2. Numerous results are described as “highly significant” (e.g., Bartlett’s test, Pillai’s trace, Wilks’ lambda), largely driven by the very large sample size. However, several effect size indicators (eigenvalues, Pillai’s trace values, Wilks’ lambda) suggest only weak to moderate discrimination. Greater emphasis should be placed on effect sizes and biological interpretation, rather than p-values alone.

3. Although deviations from normality were detected using Kolmogorov–Smirnov tests, the data were still considered approximately normal based on Q–Q plots. This subjective justification is insufficient, particularly given the reliance on CDA and CCA. The implications of heteroscedasticity for model robustness and discriminant performance are not adequately addressed.

Discussion

1. While the Discussion is comprehensive, it is considerably longer than necessary and reads more like an extended review than a focused interpretation of the study’s findings. Several paragraphs reiterate well-established concepts in dairy ruminant physiology (nutrition–fertility links, endocrine regulation, SCC and mastitis) without sufficiently anchoring them to the specific results of the present analyses. The authors should condense the Discussion, prioritizing interpretation of their own canonical and classification results rather than broad background knowledge.

2. The Discussion frequently implies causal mechanisms (e.g., milk protein or fat “indicating” or “driving” fertility outcomes) despite the study being observational and multivariate in nature. Given the reliance on CDA, RCCA, and CHAID analyses, conclusions should be framed more cautiously as associations or predictive markers, not physiological drivers. Statements implying mechanistic causality should be revised or explicitly acknowledged as hypotheses.

3. Milk protein, fat, dry matter, and lactose are repeatedly described as reliable physiological markers of fertility. However, these traits are indirect proxies influenced by multiple confounding factors (stage of lactation, parity, management, season, and health). The Discussion does not sufficiently acknowledge these limitations, nor does it discuss whether the multivariate models adequately controlled for them.

4. Several numerical thresholds (e.g., protein, fat, lactose ranges) are restated in detail, repeating information already presented in the Results. The Discussion should focus on interpretation and implications, not restatement of findings.

5. The writing is generally clear but occasionally verbose, with long sentences that reduce readability. Some paragraphs could be merged or shortened without loss of meaning.

6. The Discussion relies heavily on cattle literature to support interpretations in goats. While reasonable, species differences should be more explicitly acknowledged.

7. Claims about “strong indicators” or “clear associations” should be aligned with the reported effect sizes and discriminant strength, not statistical significance alone.

8. The final summary is well written but could be more concise and should better reflect the predictive rather than causal nature of the findings.

Conclusions

The Conclusions section is longer than necessary and overlaps substantially with the Discussion. Several statements reiterate mechanistic interpretations and management implications that have already been addressed in detail earlier in the manuscript. Conclusions should be more concise and focused strictly on the principal findings. In addition, some claims, particularly regarding integration into genetic indices, improvements in animal welfare, and the practical implementation of CHAID models for breeding decisions, are presented too conclusively given the observational and correlational nature of the analyses. The authors are encouraged to shorten this section, avoid introducing new interpretations, and phrase broader implications more cautiously, or alternatively relocate them to the Discussion or Future Perspectives.

.

Reviewer #1: **Yes:**INTAN SUHANA ZULKAFLIINTAN SUHANA ZULKAFLIINTAN SUHANA ZULKAFLIINTAN SUHANA ZULKAFLI

Reviewer #2: **Yes:**Zeayd Fadhil SaeedZeayd Fadhil SaeedZeayd Fadhil SaeedZeayd Fadhil Saeed

Reviewer #3: **Yes:**Mina Bagheri VarzanehMina Bagheri VarzanehMina Bagheri VarzanehMina Bagheri Varzaneh

Reviewer #4: No

Reviewer #5: **Yes:**Sabreena AlamSabreena AlamSabreena AlamSabreena Alam

Reviewer #6: No

Reviewer #7: No

---

## [Author Response · Author response to Decision Letter 1]

24 Mar 2026

Francisco Javier Navas González

Department of Genetics, Faculty of Veterinary Sciences

University of Córdoba

Rabanales University Campus, 14071

Córdoba (Spain)

+34 651679262

fjng87@hotmail.com

19/02/2026

Dear Editor,

All the team responsible for this paper acknowledge the comments from the reviewers and editor, as they help to improve the quality of our manuscript. In the following paragraphs, we will describe and address how referees’ new recommendations were followed. A point-by-point response to comments is provided as well as a file where changes are highlighted

Response: We thank the Editorial Office for this reminder. The manuscript has been carefully revised to comply with all PLOS ONE style and formatting requirements, including file naming conventions, manuscript structure, and layout, following the official PLOS ONE templates for the main body and for the title, authors, and affiliations. The revised files have been prepared accordingly and uploaded in the required format.

Funding was not received for the development of the present study. The present research was carried out during the covering period of a Ramón y Cajal Post-Doctoral Contract with the reference MCIN/AEI/10.13039/501100011033 and the European Union “NextGenerationEU”/PRTR.

Response: We added "The funders had no role in study design, data collection and analysis, decision to publish, or preparation of the manuscript."

Response: We moved ethics statement to the Methods section.

4. In the online submission form, you indicated that data will be made available from the corresponding author F.J.N.G. upon reasonable request.

Response: We modified the data availability statement.

Response: We modified the data availability statement.

6. We note that Figure 2 in your submission contain map images which may be copyrighted. All PLOS content is published under the Creative Commons Attribution License (CC BY 4.0), which means that the manuscript, images, and Supporting Information files will be freely available online, and any third party is permitted to access, download, copy, distribute, and use these materials in any way, even commercially, with proper attribution. For these reasons, we cannot publish previously copyrighted maps or satellite images created using proprietary data, such as Google software (Google Maps, Street View, and Earth). For more information, see our copyright guidelines: http://journals.plos.org/plosone/s/licenses-and-copyright.

“I request permission for the open-access journal PLOS ONE to publish XXX under the Creative Commons Attribution License (CCAL) CC BY 4.0

(http://creativecommons.org/licenses/by/4.0/). Please be aware that this license allows unrestricted use and distribution, even commercially, by third parties. Please reply and provide explicit written permission to publish XXX under a CC BY license and complete the attached form.”

Response: The maps included in this publication were created by the authors specifically for this study.

Response: We included them.

Reviewers' comments:

Reviewer's Responses to Questions

Comments to the Author

Review Comments to the Author

Reviewer #1: Abstract: Well written and adequate

Response: We thank the reviewer for his kind comment.

Introduction: Mostly well written. There is a paragraph on genetic influence and list of genes associated with milk production and fertility. These genes were not investigated in the current study. Suggest to trim down the genetic mention.

Response: We thank the reviewer for this constructive comment. We agree that the original version of the Introduction placed excessive emphasis on specific candidate genes that were not directly investigated in the present study. To better align the background section with the actual scope of the analyses, we have trimmed the genetic discussion by removing detailed gene lists and refocusing the paragraph on the broader role of genetic background acting through endocrine and metabolic pathways. This revision preserves the conceptual framework motivating the study while avoiding the implication that these specific genes were examined empirically.

Materials & methods:

Extensive, well-thought out & well planned methodologies.

These were conducted well and written/explained well in the methodology section.

Measures have been taken to remove/reduce bias.

Response: We thank the reviewer for this positive assessment of the methodology. Considerable effort was devoted to designing a rigorous and transparent methodological framework, and we are pleased that the clarity of the description and the measures implemented to minimize potential sources of bias were recognized. Where appropriate, explicit procedures were adopted to control for confounding effects and ensure consistency across observations, thereby strengthening the robustness and reproducibility of the analyses.

Results:

There are extensive explanation of the results which is not needed in the result section and should be better saved for the discussion.

All figures given were of high quality.

Response: We agree with the reviewer’s observation. The Results section has been revised to remove extensive interpretative explanations, which have now been transferred to the Discussion section. The Results are currently presented in a descriptive manner, focusing strictly on the observed data and statistical outputs. We appreciate the reviewer’s positive assessment regarding the quality of the figures, which have been retained unchanged.

Discussion:

The discussion is adequate. There is a tendency to mention genetic and hormonal components that play important roles in fertility and milk production at many instances, but these were not really investigated in the study. Instead, they highlighted what the study is lacking. Therefore a clear mention of study limitations and future direction would be good to supplement these “uninvestigated” aspects of milk prodution vs fertility.

Response: We thank the reviewer for this insightful comment. We agree that several genetic and hormonal mechanisms discussed were not directly investigated in the present study and were included to provide biological context rather than empirical evidence. To address this concern, we have now explicitly acknowledged these aspects as study limitations and added a dedicated paragraph in the Discussion outlining the observational nature of the data and the absence of direct hormonal, metabolic, and genetic measurements. This new paragraph also defines clear directions for future research aimed at integrating molecular, endocrine, and fertility endpoints with milk production traits. These revisions ensure that the Discussion remains balanced and that uninvestigated mechanisms are clearly framed as perspectives rather than conclusions.

Reviewer #2: . 1- Strengths (Positive Points)

A. Strong and Highly Relevant Research Topic

- The manuscript addresses a significant challenge in dairy small ruminants: the balance between milk production and fertility, particularly in Murciano-Granadina goats, an economically important breed.

- The focus on canonical pathways linking production and fertility is innovative and aligns with current priorities in animal breeding and reproductive physiology.

B. Extremely Large and Valuable Dataset

- The study includes 32,693 insemination records and 29,390 milk composition records, covering over 21,000 goats.

- This exceptional sample size strengthens the generalizability and statistical power of the results.

C. Advanced Statistical and Multivariate Methods

- The authors employ a combination of Canonical Discriminant Analysis (CDA), Regularized Canonical Correlation Analysis (RCCA), CHAID decision trees, and multicollinearity filtering using VIF.

- This multilevel analytical framework is sophisticated and appropriate for revealing structure in complex biological relationships.

D. Clear Identification of Physiological and Endocrine Mechanisms

- The manuscript provides a comprehensive background on:

- Endocrine pathways (IGF-1, leptin, HPG axis).

- Genetic polymorphisms (IGF1, LEP, DGAT1, etc.).

- Physiological balance between production and reproduction.

- This supports the biological interpretation of the statistical findings.

E. Strong Practical Implications

- The study identifies specific thresholds for milk traits associated with optimal fertility (e.g., protein 3.64–6.98%, fat 6.0–7.9%, SCC < 5200×10³).

- Results have direct value for breeding programs, AI centers, and dairy goat farmers.

F. Data Quality Control

The authors conduct:

Outlier screening (ROUT),

Normality checks (Kolmogorov–Smirnov),

Homoscedasticity testing (Levene),

Multicollinearity evaluation (VIF),

Cross-validation.

These steps confirm good methodological rigor.

Response: We sincerely thank Reviewer #2 for this thorough and positive evaluation of our manuscript. We greatly appreciate the recognition of the relevance of the research topic, the exceptional size and value of the dataset, and the robustness of the multivariate analytical framework employed. We are also grateful for the reviewer’s acknowledgment of the biological context provided for interpreting the statistical results, as well as the practical implications of the identified milk trait thresholds for fertility in Murciano-Granadina goats. Finally, we appreciate the reviewer’s attention to the data quality control procedures applied, which we consider essential for ensuring the reliability and interpretability of the findings. These encouraging comments confirm the scientific and applied value of the study and have been very helpful in refining the revised version of the manuscript.

2. Weaknesses (Negative Points)

A. Manuscript Organization and Length

- The manuscript is unusually long, especially the Materials and Methods section, which contains:

- Excessive detail on AI protocols,

- Geography of farms,

- Temperature and climatic descriptions,

- Very long mathematical formulas.

- Some methodological details could be moved to Supplementary Materials to improve readability.

Response: We thank the reviewer for this observation. To address concerns regarding manuscript length and level of detail, the Materials and Methods section was substantially condensed, with its total length reduced by approximately one third (from 3,180 to 2,383 words).

• The artificial insemination (AI) protocol section was shortened (from 183 to 161 words) while retaining all essential methodological information.

• Descriptions of farm geography and climatic and

---

## [Editor Report · Decision Letter 1]

14 Apr 2026

A comprehensive analysis of canonical biological pathways linking milk yield and quality traits to key fertility indicators in Murciano-Granadina dairy does

PONE-D-25-64731R1

Dear Dr. Navas González,

We’re pleased to inform you that your manuscript has been judged scientifically suitable for publication and will be formally accepted for publication once it meets all outstanding technical requirements.

Kind regards,

Ghadeer Sabah Bustani, Ph.D

Academic Editor

PLOS One
---

## [Editor Report · Acceptance letter]

PONE-D-25-64731R1

PLOS One

Dear Dr. Navas González,

I'm pleased to inform you that your manuscript has been deemed suitable for publication in PLOS One. Congratulations! Your manuscript is now being handed over to our production team.

Kind regards,

on behalf of

Dr. Ghadeer Sabah Bustani

Academic Editor

PLOS One